# Positive-unlabeled AUC Maximization under Covariate Shift

**Atsutoshi Kumagai** [1]   **Tomoharu Iwata** [1]   **Hiroshi Takahashi** [1]
**Taishi Nishiyama** [1]   **Kazuki Adachi** [1,2]   **Yasuhiro Fujiwara** [1]

## Abstract

Maximizing the area under the receiver operating characteristic curve (AUC) is a standard approach to imbalanced binary classification tasks. Existing AUC maximization methods typically assume that training and test distributions are identical. However, this assumption is often violated due to *a covariate shift*, where the input distribution can vary but the conditional distribution of the class label given the input remains unchanged. The importance weighting is a common approach to the covariate shift, which minimizes the test risk with importance-weighted training data. However, it cannot maximize the AUC. In this paper, to achieve this, we theoretically derive two estimators of the test AUC risk under the covariate shift by using positive and unlabeled (PU) data in the training distribution and unlabeled data in the test distribution. Our first estimator is calculated from importance-weighted PU data in the training distribution, and the second one is calculated from importance-weighted positive data in the training distribution and unlabeled data in the test distribution. We train classifiers by minimizing a weighted sum of the two AUC risk estimators that is approximately equivalent to the test AUC risk. Unlike the existing importance weighting, our method does not require negative labels and class-priors. We experimentally show the effectiveness of our method with six real-world datasets.

## 1. Introduction

In many real-world binary classification tasks such as cyber security (Mirsky et al., 2018; Bagui & Li, 2021), medical care (Yang et al., 2021), and product inspection (Park et al., 2016), *class-imbalance* often occurs where positive data is much smaller than negative data (Johnson & Khoshgoftaar, 2019). In this situation, classification accuracy, which is the standard performance measure for ordinary classification, is not a suitable measure (Ueda & Fujino, 2018; Yang & Ying, 2022). Instead, the area under the receiver operating characteristic curve (AUC) is widely used (Bradley, 1997; Huang & Ling, 2005; McDermott et al., 2024). The AUC is the probability that a classifier will rank a randomly chosen positive instance higher than a randomly chosen negative one (Yang & Ying, 2022). Thanks to the nature of the ranking, the AUC can adequately measure the classifier's performance even with imbalanced data. By maximizing the AUC, we can learn accurate classifiers from imbalanced data (Brefeld et al., 2005; Yang & Ying, 2022; Ying et al., 2016; Liu et al., 2020; Yuan et al., 2021a).

Existing AUC maximization methods usually assume that training and test distributions are identical. However, this assumption is often violated in practice due to distribution shifts. This paper considers *a covariate shift*, where the input distribution can vary but the conditional distribution of the class label given the input remains unchanged between training and test stages (Shimodaira, 2000). The covariate shift is the most common distribution shift (He et al., 2023) and often occurs in imbalanced classification tasks. For example, in medical care, the distribution of patients (input instances) can vary due to the differences in hospitals and measurement instruments, even when the conditional distribution of the class label given the input does not change (Matsui et al., 2019). In cyber security, attackers rapidly generate new attacks (input instances), and thus the input distribution can vary over time (Kumagai & Iwata, 2016).

When labeled positive and negative data in the training distribution and unlabeled data in the test distribution are available, the covariate shift on ordinary classification can be alleviated using the widely used importance weighting framework (Sugiyama et al., 2012; Lu et al., 2022). It first estimates importance weights that are the ratio between training and test input densities and then learns classifiers by minimizing the importance-weighted empirical training risk that is approximately equivalent to the test risk. However, it is not designed for AUC maximization. In addition, labeled negative data in the training distribution are often difficult to collect in some applications. For example, in

---

[1]NTT Corporation, Japan [2]Yokohama National University, Japan. Correspondence to: Atsutoshi Kumagai <atsutoshi.kumagai@ntt.com>.

*Proceedings of the 42ⁿᵈ International Conference on Machine Learning*, Vancouver, Canada. PMLR 267, 2025. Copyright 2025 by the author(s).

cyber security, although some malicious data (positive data) can be collected from public sources such as blocklists, benign data (negative data) are often unavailable due to privacy reasons, and identifying benign data within given unlabeled data requires a high level of expertise (Mirsky et al., 2018; Sharafaldin et al., 2018). Although an importance weighting method (Sakai & Shimizu, 2019) does not use labeled negative data exceptionally, it cannot maximize the AUC.

In this paper, we propose a method for maximizing the AUC under the covariate shift by using labeled positive and unlabeled (PU) data in the training distribution and unlabeled data in the test distribution. On the basis of the importance weighting, we theoretically derive two estimators of the AUC risk on the test distribution. The first one is calculated from importance-weighted PU data in the training distribution. While this estimator is effective, unlabeled data in the test distribution are only used for estimating importance weights and are not directly used in classifier learning, even if they have useful information. This drawback is common in ordinary importance weighting methods (Sugiyama & Kawanabe, 2012; Fang et al., 2020; Sakai & Shimizu, 2019). The second estimator is calculated from importance-weighted positive data in the training distribution and unlabeled data in the test distribution. This estimator directly uses unlabeled data in the test distribution for classifier learning. Our loss function for classifier learning is a weighted sum of these two AUC risk estimators that is also approximately equivalent to the test AUC risk. By using this loss, we can directly use all available data in our setting for classifier learning. Moreover, unlike the existing method (Sakai & Shimizu, 2019), our method does not require class-priors for training, which is beneficial since they are generally difficult or impossible to estimate (Zhao et al., 2023; Yao et al., 2021).

Although we can learn classifiers by minimizing the loss, the importance weights are difficult to estimate, especially when using complex models such as neural networks or complex data such as image data (Fang et al., 2020; Kato & Teshima, 2021; Rhodes et al., 2020). Thus, the above two-step importance weighting often does not work well in such cases. To deal with this problem, following recent works (Fang et al., 2020; 2023), we use a dynamic approach that iterates the importance weight estimation and classifier learning while sharing a neural network for feature extraction. By training the shared feature extractor with simpler classifier learning, the importance weights can be estimated more easily; the classifier learning can be performed without biases by using the estimated importance weights.

Our main contributions are as follows:

- We propose a novel and practical problem setting, where the aim is to maximize the AUC under the covariate shift with PU data in the training distribution

and unlabeled data in the test distribution.

- We theoretically derive two estimators of the test AUC risk under our problem setting, which can be used for classifier learning without class-priors.

- We develop a dynamic approach for the importance weighting with the derived AUC risk estimators.

- We experimentally show that the proposed method outperforms various existing methods with real-world datasets.

## 2. Related Work

Many AUC maximization methods have been proposed (Brefeld et al., 2005; Yang & Ying, 2022; Ying et al., 2016; Liu et al., 2020; Yuan et al., 2021a). As reported in previous studies (Yuan et al., 2021a;b; Fujino & Ueda, 2016; Wang et al., 2023), they often outperform other methods for imbalanced classification such as class balanced loss (Charoenphakdee et al., 2019), focal loss (Lin et al., 2017), or sampling-based methods (Chawla et al., 2002; Menardi & Torelli, 2014). However, these AUC maximization methods require labeled positive and negative data and assume that the training and test distributions are identical. Thus, they are inappropriate for our problem setting where there are no labeled negative data and the covariate shift occurs.

Covariate shift adaptation methods attempt to learn accurate models under the covariate shift by using unlabeled data in the test distribution and labeled data in the training distribution (Pan & Yang, 2009; Sugiyama & Kawanabe, 2012). The importance weighting is a representative approach for the covariate shift (Sugiyama & Kawanabe, 2012). This approach first estimates importance weights and then minimizes the importance-weighted empirical training risk. Although this two-step approach is theoretically sound, it often does not work well for complex models or data (Fang et al., 2020; 2023). To overcome this difficulty, Fang et al. (2020; 2023) have recently proposed a dynamic approach that iterates importance weight estimation and classifier learning, which enables the importance weighting to work well in such difficult cases. Our method also uses the dynamic approach. Another approach is to learn invariant feature representations by minimizing the discrepancy of the features between the training and test distributions (Long et al., 2015; Sun et al., 2017; 2016; Ganin & Lempitsky, 2015; Kumagai & Iwata, 2019; Shen et al., 2018). Although this approach is promising, it often deteriorates the performance since it does not explicitly minimize the test risk (Zhao et al., 2019; Kumagai et al., 2024). These existing methods usually minimize the classification risk (or negative classification accuracy), which is an inappropriate metric for imbalanced data. One distribution adaptation method has been proposed to maximize the AUC by learning invariant

feature representations (Yang et al., 2023). However, all these methods including the method (Yang et al., 2023) require positive and negative data in the training distribution, which are unavailable in our problem setting.

PU learning methods aim to learn binary classifiers by using only PU data (Bekker & Davis, 2020). Our work is closely related to PU learning since it assumes PU data in the training distribution. A representative approach for PU learning is the empirical risk minimization approach, which rewrites the empirical classification risk by using only PU data (Du Plessis et al., 2015; Kiryo et al., 2017; Sugiyama et al., 2022; Jiang et al., 2023). Although they are effective, they cannot maximize the AUC. Recent studies have shown that the AUC can be maximized from only PU data by using the techniques of PU learning (Sakai et al., 2018; Xie & Li, 2018; Charoenphakdee et al., 2019; Xie et al., 2024). However, these methods assume that training and test distributions are the same.

Several PU learning methods for distribution shift have been proposed that use unlabeled data in the test distribution and PU data in the training distribution (Sakai & Shimizu, 2019; Kumagai et al., 2024; Nakajima & Sugiyama, 2023). Nakajima and Sugiyama (2023) considers a class-prior shit, where the class-prior can vary, but their method cannot treat the covariate shift and maximize the AUC. Although Sakai and Shimizu (2019) consider the covariate shift, they treat the classification risk and cannot maximize the AUC. Due to the pairwise formulation of the AUC, this method using ordinary instance-wise loss functions cannot be straightforwardly applied to the AUC. Kumagai et al. (2024) maximize the AUC under a positive distribution shift, where negative-conditional density does not change but positive-conditional density can vary. This method, however, cannot deal with the covariate shift that often occurs in practice. Additionally, these methods assume that the class-prior on the training distribution is available, which is generally difficult to estimate (Yao et al., 2021). In contrast, the proposed method does not require it.

## 3. Preliminary

We briefly explain AUC maximization. Let input instance $\mathbf{x} \in \mathcal{X}$ and its class label $y \in \{-1, +1\}$ be equipped with probability density $p(\mathbf{x}, y)$, where $+1$ and $-1$ mean a positive and negative class, respectively. $p^{\mathrm{p}}(\mathbf{x}) := p(\mathbf{x}|y = +1)$ and $p^{\mathrm{n}}(\mathbf{x}) := p(\mathbf{x}|y = -1)$ are the conditional probability densities of positive and negative classes, respectively. Let $s : \mathcal{X} \to \mathbb{R}$ be a score function that outputs the positivity of an input instance. The classifier is defined by the score function with threshold $t$: $y = \mathrm{sign}(s(\mathbf{x}) - t)$, where sign is a sign function.

The AUC is the probability of a randomly drawn positive instance being ranked before a randomly drawn negative instance (Yang & Ying, 2022). Specifically, the AUC with score function $s$ can be formulated as

$$\mathrm{AUC}(s) = \mathbb{E}_{\mathbf{x}^{\mathrm{p}} \sim p^{\mathrm{p}}(\mathbf{x})} \mathbb{E}_{\mathbf{x}^{\mathrm{n}} \sim p^{\mathrm{n}}(\mathbf{x})} \left[ I(s(\mathbf{x}^{\mathrm{p}}) \geq s(\mathbf{x}^{\mathrm{n}})) \right]$$
$$= 1 - \mathbb{E}_{\mathbf{x}^{\mathrm{p}} \sim p^{\mathrm{p}}(\mathbf{x})} \mathbb{E}_{\mathbf{x}^{\mathrm{n}} \sim p^{\mathrm{n}}(\mathbf{x})} \left[ I(s(\mathbf{x}^{\mathrm{p}}) < s(\mathbf{x}^{\mathrm{n}})) \right], \quad (1)$$

where $I(z)$ is the indicator function that outputs $1$ if $z$ is true and $0$ otherwise, and $\mathbb{E}$ is the expectation. Maximizing the AUC is equivalent to minimizing the AUC risk,

$$\mathcal{R}(s) := \mathbb{E}_{\mathbf{x}^{\mathrm{p}} \sim p^{\mathrm{p}}(\mathbf{x})} \mathbb{E}_{\mathbf{x}^{\mathrm{n}} \sim p^{\mathrm{n}}(\mathbf{x})} \left[ I(s(\mathbf{x}^{\mathrm{p}}) < s(\mathbf{x}^{\mathrm{n}})) \right]. \quad (2)$$

Since the gradient of indicator function $I$ is zero everywhere except for the origin, the AUC risk cannot be minimized via gradient descent methods. To avoid this, the following smoothed AUC risk is often used by replacing the indicator function with a sigmoid function $\sigma(z) = 1/(1 + \exp(-z))$ (Iwata & Yamanaka, 2019; Kumagai et al., 2019; 2024):

$$\mathcal{R}_{\sigma}(s) := \mathbb{E}_{\mathbf{x}^{\mathrm{p}} \sim p^{\mathrm{p}}(\mathbf{x})} \mathbb{E}_{\mathbf{x}^{\mathrm{n}} \sim p^{\mathrm{n}}(\mathbf{x})} \left[ \sigma(-s(\mathbf{x}^{\mathrm{p}}) + s(\mathbf{x}^{\mathrm{n}})) \right]. \quad (3)$$

Given $N^{\mathrm{p}}$ positive instances $\{\mathbf{x}_1^{\mathrm{p}}, \ldots, \mathbf{x}_{N^{\mathrm{p}}}^{\mathrm{p}}\}$ drawn from $p^{\mathrm{p}}(\mathbf{x})$ and $N^{\mathrm{n}}$ negative instances $\{\mathbf{x}_1^{\mathrm{n}}, \ldots, \mathbf{x}_{N^{\mathrm{n}}}^{\mathrm{n}}\}$ drawn from $p^{\mathrm{n}}(\mathbf{x})$, the empirical estimator of the smoothed AUC risk is calculated as

$$\widehat{\mathcal{R}}_{\sigma}(s) = \frac{1}{N^{\mathrm{p}} N^{\mathrm{n}}} \sum_{n=1}^{N^{\mathrm{p}}} \sum_{m=1}^{N^{\mathrm{n}}} \left[ \sigma(-s(\mathbf{x}_n^{\mathrm{p}}) + s(\mathbf{x}_m^{\mathrm{n}})) \right]. \quad (4)$$

By minimizing this empirical smoothed AUC risk w.r.t. the parameters of $s$, we can obtain good score functions to maximize the AUC when the training and test distributions are identical (Yang & Ying, 2022).

## 4. Proposed Method

In this section, we first explain our problem setting (subsection 4.1). Then, we theoretically derive two estimators of the AUC risk on the test distribution under the covariate shift and define our loss function for classifier learning on the basis of the derived two AUC risk estimators (subsection 4.2). After describing the importance weight estimation method (subsection 4.3), we finally explain our training procedure (subsection 4.4).

### 4.1. Problem Setting

Suppose that we are given a set of positive instances $X_{\mathrm{tr}}^{\mathrm{p}}$ and a set of unlabeled instances $X_{\mathrm{tr}}$ drawn from the training distribution:

$$X_{\mathrm{tr}}^{\mathrm{p}} = \{\mathbf{x}_{\mathrm{tr},n}^{\mathrm{p}}\}_{n=1}^{N_{\mathrm{tr}}^{\mathrm{p}}} \sim p_{\mathrm{tr}}^{\mathrm{p}}(\mathbf{x}) := p_{\mathrm{tr}}(\mathbf{x}|y = +1), \quad (5)$$

$$X_{\mathrm{tr}} = \{\mathbf{x}_{\mathrm{tr},n}\}_{n=1}^{N_{\mathrm{tr}}} \sim p_{\mathrm{tr}}(\mathbf{x}) = \pi_{\mathrm{tr}} p_{\mathrm{tr}}^{\mathrm{p}}(\mathbf{x}) + (1 - \pi_{\mathrm{tr}}) p_{\mathrm{tr}}^{\mathrm{n}}(\mathbf{x}), \quad (6)$$

where $p_{\text{tr}}(\mathbf{x})$ is the marginal density of the training distribution, $p_{\text{tr}}^{\text{p}}(\mathbf{x})$ and $p_{\text{tr}}^{\text{n}}(\mathbf{x}) := p_{\text{tr}}(\mathbf{x}|y = -1)$ are positive- and negative-conditional densities of the training distribution, respectively, and $\pi_{\text{tr}} := p_{\text{tr}}(y = +1)$ is the positive class-prior. We also suppose that a set of unlabeled instances $X_{\text{te}}$ drawn from the test distribution is given:

$$X_{\text{te}} = \{\mathbf{x}_{\text{te},n}\}_{n=1}^{N_{\text{te}}} \sim p_{\text{te}}(\mathbf{x}) = \pi_{\text{te}} p_{\text{te}}^{\text{p}}(\mathbf{x}) + (1 - \pi_{\text{te}}) p_{\text{te}}^{\text{n}}(\mathbf{x}),$$
(7)

where $p_{\text{te}}^{\text{p}}(\mathbf{x}) := p_{\text{te}}(\mathbf{x}|y = +1)$, $p_{\text{te}}^{\text{n}}(\mathbf{x}) := p_{\text{te}}(\mathbf{x}|y = -1)$, and $\pi_{\text{te}} := p_{\text{te}}(y = +1)$. As shown later, we do not need to know specific values of $\pi_{\text{tr}}$ and $\pi_{\text{te}}$ to maximize the AUC [1], which is beneficial in practice.

We consider the covariate shift between the training and test distributions, where the marginal density can vary but the conditional density of the class label given the input instance remains unchanged,

$$p_{\text{tr}}(\mathbf{x}) \neq p_{\text{te}}(\mathbf{x}), \ \ p_{\text{tr}}(y|\mathbf{x}) = p_{\text{te}}(y|\mathbf{x}) =: p(y|\mathbf{x}). \quad (8)$$

This situation often occurs in imbalanced classification tasks as described in Section 1. Our goal is to learn the score function $s : \mathcal{X} \to \mathbb{R}$ that can maximize the AUC on the test distribution by using $X_{\text{tr}}^{\text{P}} \cup X_{\text{tr}} \cup X_{\text{te}}$. In the following, $C$ represents any constant that does not depend on models to be learned (e.g., score function $s$).

### 4.2. Importance-weighted AUC Risks

In this subsection, we theoretically derive two estimators of the AUC risk on the test distribution under the covariate shift. The objective function to be minimized is the following smoothed AUC risk on the test distribution,

$$\mathcal{R}_\sigma^{\text{te}}(s) := \mathbb{E}_{\mathbf{x}^{\text{p}} \sim p_{\text{te}}^{\text{p}}(\mathbf{x})} \mathbb{E}_{\mathbf{x}^{\text{n}} \sim p_{\text{te}}^{\text{n}}(\mathbf{x})} [f(\mathbf{x}^{\text{p}}, \mathbf{x}^{\text{n}})], \quad (9)$$

where we set $f(\mathbf{x}^{\text{p}}, \mathbf{x}^{\text{n}}) := \sigma(-s(\mathbf{x}^{\text{p}}) + s(\mathbf{x}^{\text{n}}))$. Since this AUC risk depends on $p_{\text{te}}^{\text{p}}(\mathbf{x})$ and $p_{\text{te}}^{\text{n}}(\mathbf{x})$, it seems to be impossible to calculate directly in our setting. However, by using the importance weighting framework, this AUC risk can be calculated as described below.

**First AUC Risk Estimator** We derive the AUC risk estimator that is calculated from PU data in the training distribution. First, positive-conditional density $p_{\text{te}}^{\text{p}}(\mathbf{x})$ can be rewritten as

$$
\begin{aligned}
p_{\text{te}}^{\text{p}}(\mathbf{x}) &= \frac{p(y = +1|\mathbf{x}) p_{\text{te}}(\mathbf{x})}{\pi_{\text{te}}} \\
&= \frac{p(y = +1|\mathbf{x}) p_{\text{tr}}(\mathbf{x})}{\pi_{\text{tr}}} \frac{p_{\text{te}}(\mathbf{x})}{p_{\text{tr}}(\mathbf{x})} \frac{\pi_{\text{tr}}}{\pi_{\text{te}}} \\
&= p_{\text{tr}}^{\text{p}}(\mathbf{x}) \frac{p_{\text{te}}(\mathbf{x})}{p_{\text{tr}}(\mathbf{x})} \frac{\pi_{\text{tr}}}{\pi_{\text{te}}} = p_{\text{tr}}^{\text{p}}(\mathbf{x}) w(\mathbf{x}) \frac{\pi_{\text{tr}}}{\pi_{\text{te}}}, \quad (10)
\end{aligned}
$$

---

[1] We only require $\pi_{\text{tr}}, \pi_{\text{te}} \in (0, 1)$ to evade zero-division when deriving our AUC risk estimators in subsection 4.2.

where we used the Bayes' theorem and the assumption of the covariate shift (Eq. (8)) in the first and third equalities, and $w(\mathbf{x}) := \frac{p_{\text{te}}(\mathbf{x})}{p_{\text{tr}}(\mathbf{x})}$ is referred to as the importance weight. Similarly, negative-conditional density $p_{\text{te}}^{\text{n}}(\mathbf{x})$ can be represented as

$$p_{\text{te}}^{\text{n}}(\mathbf{x}) = p_{\text{tr}}^{\text{n}}(\mathbf{x}) w(\mathbf{x}) \frac{1 - \pi_{\text{tr}}}{1 - \pi_{\text{te}}}. \quad (11)$$

By substituting Eqs. (10) and (11) into Eq. (9), we can obtain

$$
\begin{aligned}
\mathcal{R}_\sigma^{\text{te}}(s) &= \frac{\pi_{\text{tr}}(1 - \pi_{\text{tr}})}{\pi_{\text{te}}(1 - \pi_{\text{te}})} \\
&\times \mathbb{E}_{\mathbf{x}^{\text{p}} \sim p_{\text{tr}}^{\text{p}}(\mathbf{x})} \mathbb{E}_{\mathbf{x}^{\text{n}} \sim p_{\text{tr}}^{\text{n}}(\mathbf{x})} [w(\mathbf{x}^{\text{p}}) w(\mathbf{x}^{\text{n}}) f(\mathbf{x}^{\text{p}}, \mathbf{x}^{\text{n}})]. \quad (12)
\end{aligned}
$$

From the definition of marginal density $p_{\text{tr}}(\mathbf{x})$ in Eq. (6), negative-conditional density $p_{\text{tr}}^{\text{n}}(\mathbf{x})$ can be expressed as

$$p_{\text{tr}}^{\text{n}}(\mathbf{x}) = \frac{1}{1 - \pi_{\text{tr}}} [p_{\text{tr}}(\mathbf{x}) - \pi_{\text{tr}} p_{\text{tr}}^{\text{p}}(\mathbf{x})]. \quad (13)$$

By substituting Eq. (13) into Eq. (12), we can obtain

$$
\begin{aligned}
\mathcal{R}_\sigma^{\text{te}}(s) =& \\
\frac{\pi_{\text{tr}}}{\pi_{\text{te}}(1 - \pi_{\text{te}})} &\mathbb{E}_{\mathbf{x}^{\text{p}} \sim p_{\text{tr}}^{\text{p}}(\mathbf{x})} \mathbb{E}_{\mathbf{x} \sim p_{\text{tr}}(\mathbf{x})} [w(\mathbf{x}^{\text{p}}) w(\mathbf{x}) f(\mathbf{x}^{\text{p}}, \mathbf{x})] \\
-\frac{\pi_{\text{tr}}^2}{\pi_{\text{te}}(1 - \pi_{\text{te}})} &\mathbb{E}_{\mathbf{x}^{\text{p}} \sim p_{\text{tr}}^{\text{p}}(\mathbf{x})} \mathbb{E}_{\bar{\mathbf{x}}^{\text{p}} \sim p_{\text{tr}}^{\text{p}}(\mathbf{x})} [w(\mathbf{x}^{\text{p}}) w(\bar{\mathbf{x}}^{\text{p}}) f(\mathbf{x}^{\text{p}}, \bar{\mathbf{x}}^{\text{p}})].
\end{aligned}
$$
(14)

Here, since $\sigma(z) + \sigma(-z) = 1$ for all $z \in \mathbb{R}$, the second term of Eq. (14) becomes

$$
\begin{aligned}
\mathbb{E}_{\mathbf{x}^{\text{p}} \sim p_{\text{tr}}^{\text{p}}(\mathbf{x})} \mathbb{E}_{\bar{\mathbf{x}}^{\text{p}} \sim p_{\text{tr}}^{\text{p}}(\mathbf{x})} &[w(\mathbf{x}^{\text{p}}) w(\bar{\mathbf{x}}^{\text{p}}) f(\mathbf{x}^{\text{p}}, \bar{\mathbf{x}}^{\text{p}})] = \\
\frac{1}{2} \mathbb{E}_{\mathbf{x}^{\text{p}} \sim p_{\text{tr}}^{\text{p}}(\mathbf{x})} &\mathbb{E}_{\bar{\mathbf{x}}^{\text{p}} \sim p_{\text{tr}}^{\text{p}}(\mathbf{x})} [w(\mathbf{x}^{\text{p}}) w(\bar{\mathbf{x}}^{\text{p}})] = C, \quad (15)
\end{aligned}
$$

where $C$ represents a constant that does not depend on $s$. Therefore, the AUC risk on the test distribution in Eq. (14) can be represented as

$$
\begin{aligned}
\mathcal{R}_\sigma^{\text{te}}(s) =& \frac{\pi_{\text{tr}}}{\pi_{\text{te}}(1 - \pi_{\text{te}})} \\
&\times \mathbb{E}_{\mathbf{x}^{\text{p}} \sim p_{\text{tr}}^{\text{p}}(\mathbf{x})} \mathbb{E}_{\mathbf{x} \sim p_{\text{tr}}(\mathbf{x})} [w(\mathbf{x}^{\text{p}}) w(\mathbf{x}) f(\mathbf{x}^{\text{p}}, \mathbf{x})] + C. \quad (16)
\end{aligned}
$$

Since this AUC risk depends on positive-conditional training density $p_{\text{tr}}^{\text{p}}(\mathbf{x})$ and marginal training density $p_{\text{tr}}(\mathbf{x})$, we can obtain the following empirical AUC risk estimator,

$$
\begin{aligned}
\widehat{\mathcal{R}}_{\sigma,1}^{\text{te}}(s) :=& \frac{\pi_{\text{tr}}}{\pi_{\text{te}}(1 - \pi_{\text{te}})} \frac{1}{N_{\text{tr}}^{\text{P}} N_{\text{tr}}} \\
&\times \sum_{n=1}^{N_{\text{tr}}^{\text{P}}} \sum_{m=1}^{N_{\text{tr}}} [w(\mathbf{x}_{\text{tr},n}^{\text{P}}) w(\mathbf{x}_{\text{tr},m}) f(\mathbf{x}_{\text{tr},n}^{\text{P}}, \mathbf{x}_{\text{tr},m})] + C.
\end{aligned}
$$
(17)

In this estimator, training instances with higher importance weights have a greater influence. The estimation method for importance weight $w(\mathbf{x})$ is described in subsection 4.3.

**Second AUC Risk Estimator** In Eq. (17), unlabeled data in the test distribution $X_{\text{te}}$ are not directly used even if they have useful information for learning $s$, although they are used for estimating importance weights as described later. This drawback is common in existing importance weighting methods (Sugiyama & Kawanabe, 2012; Fang et al., 2020; Sakai & Shimizu, 2019). To deal with this problem, we derive another AUC risk estimator that is calculated from positive data in the training distribution and unlabeled data in the test distribution. Specifically, since $w(\mathbf{x}) = p_{\text{te}}(\mathbf{x})/p_{\text{tr}}(\mathbf{x})$, the equation $\mathbb{E}_{\mathbf{x} \sim p_{\text{tr}}(\mathbf{x})}[w(\mathbf{x})g(\mathbf{x})] = \mathbb{E}_{\mathbf{x} \sim p_{\text{te}}(\mathbf{x})}[g(\mathbf{x})]$ is satisfied for any function $g$. Therefore, Eq. (16) can be rewritten as

$$\mathcal{R}_\sigma^{\text{te}}(s) = \frac{\pi_{\text{tr}}}{\pi_{\text{te}}(1 - \pi_{\text{te}})}$$
$$\times \mathbb{E}_{\mathbf{x}^{\text{P}} \sim p_{\text{tr}}^{\text{P}}(\mathbf{x})} \mathbb{E}_{\mathbf{x} \sim p_{\text{te}}(\mathbf{x})}[w(\mathbf{x}^{\text{P}})f(\mathbf{x}^{\text{P}}, \mathbf{x})] + C. \quad (18)$$

This AUC risk depends on positive-conditional training density $p_{\text{tr}}^{\text{P}}(\mathbf{x})$ and marginal test density $p_{\text{te}}(\mathbf{x})$. Thus, we can obtain the following empirical AUC risk estimator,

$$\widehat{\mathcal{R}}_{\sigma,2}^{\text{te}}(s) := \frac{\pi_{\text{tr}}}{\pi_{\text{te}}(1-\pi_{\text{te}})} \frac{1}{N_{\text{tr}}^{\text{P}} N_{\text{te}}}$$
$$\times \sum_{n=1}^{N_{\text{tr}}^{\text{P}}} \sum_{m=1}^{N_{\text{te}}} \left[ w(\mathbf{x}_{\text{tr},n}^{\text{P}}) f(\mathbf{x}_{\text{tr},n}^{\text{P}}, \mathbf{x}_{\text{te},m}) \right] + C. \quad (19)$$

Since Eqs. (17) and (19) do not explicitly depend on $X_{\text{te}}$ and $X_{\text{tr}}$ except for importance weights, respectively, we consider using both simultaneously for learning $s$.

**Our Loss Function for Classifier Learning** Our loss function for learning $s$ is a weighted sum of two estimators of the AUC risk in Eqs. (17) and (19),

$$\widehat{\mathcal{R}}_\sigma^{\text{te}}(s) := \beta \widehat{\mathcal{R}}_{\sigma,1}^{\text{te}}(s) + (1 - \beta)\widehat{\mathcal{R}}_{\sigma,2}^{\text{te}}(s) = \frac{\pi_{\text{tr}}}{\pi_{\text{te}}(1-\pi_{\text{te}})}$$
$$\times \left[ \frac{\beta}{N_{\text{tr}}^{\text{P}} N_{\text{tr}}} \sum_{n=1}^{N_{\text{tr}}^{\text{P}}} \sum_{m=1}^{N_{\text{tr}}} \left[ w(\mathbf{x}_{\text{tr},n}^{\text{P}}) w(\mathbf{x}_{\text{tr},m}) f(\mathbf{x}_{\text{tr},n}^{\text{P}}, \mathbf{x}_{\text{tr},m}) \right] \right.$$
$$\left. + \frac{1 - \beta}{N_{\text{tr}}^{\text{P}} N_{\text{te}}} \sum_{n=1}^{N_{\text{tr}}^{\text{P}}} \sum_{m=1}^{N_{\text{te}}} \left[ w(\mathbf{x}_{\text{tr},n}^{\text{P}}) f(\mathbf{x}_{\text{tr},n}^{\text{P}}, \mathbf{x}_{\text{te},m}) \right] \right] + C, \quad (20)$$

where $\beta \in [0, 1]$ is a weighting hyperparameter. This loss directly uses all data $X_{\text{tr}}^{\text{P}} \cup X_{\text{tr}} \cup X_{\text{te}}$ for learning $s$. Since coefficient $\frac{\pi_{\text{tr}}}{\pi_{\text{te}}(1-\pi_{\text{te}})}$ and constant $C$ do not affect the optimization for $s$, we can safely ignore them for training. This loss without the coefficient does not depend on class-priors $\pi_{\text{tr}}$ and $\pi_{\text{te}}$, which is beneficial in practice since they are generally difficult or impossible to estimate (Zhao et al., 2023). Note that although we use the sigmoid function in the AUC risk in Eq. (3), when we use symmetric functions (i.e., function $\sigma$ satisfying $\sigma(z) + \sigma(-z) = k$ for any $z \in \mathbb{R}$

and $k$ is a constant), we can derive the loss function of the same form in Eq. (20). This is because the second term in Eq. (14) also becomes constant. The symmetric functions include many popular functions such as sigmoid, ramp, and unhinged functions (Charoenphakdee et al., 2019).

### 4.3. Importance Weight Estimation

In Eq. (20), importance weight $w(\mathbf{x}) = \frac{p_{\text{te}}(\mathbf{x})}{p_{\text{tr}}(\mathbf{x})}$ is unknown. A common way for estimating importance weights is to use density-ratio estimation methods that directly estimate the ratio between training and test densities from data without density estimation (Sugiyama et al., 2012; Kanamori et al., 2009; Huang et al., 2006). However, they are known to be unstable since $w(\mathbf{x})$ is unbounded, i.e., it takes extremely larger values around a low-density region of the denominator (Yamada et al., 2013). To alleviate this problem, we use the following relative density-ratio as the importance weight,

$$w_\alpha(\mathbf{x}) := \frac{p_{\text{te}}(\mathbf{x})}{\alpha p_{\text{te}}(\mathbf{x}) + (1 - \alpha)p_{\text{tr}}(\mathbf{x})}, \quad (21)$$

where $\alpha \in [0, 1]$ is a hyperparameter (Yamada et al., 2013). $w_\alpha(\mathbf{x})$ is bounded above by $1/\alpha$, and it is equivalent to $w(\mathbf{x})$ when $\alpha = 0$. Thus, $w_\alpha(\mathbf{x})$ is a bounded extension of $w(\mathbf{x})$. The importance weighting with the relative density-ratio has been reported to perform excellently in various data (Yamada et al., 2013; Sakai & Shimizu, 2019).

Let $r(\mathbf{x}) \in [0, 1/\alpha]$ be a model such as a neural network for estimating $w_\alpha(\mathbf{x})$. To learn parameters of $r$, we minimize the expected squared error between true importance weight $w_\alpha(\mathbf{x})$ and $r(\mathbf{x})$ as in the previous studies (Yamada et al., 2013; Kanamori et al., 2009):

$$J(r) := \mathbb{E}_{p_\alpha(\mathbf{x})} \left[ (w_\alpha(\mathbf{x}) - r(\mathbf{x}))^2 \right]$$
$$= \mathbb{E}_{p_{\text{te}}(\mathbf{x})} \left[ \alpha r(\mathbf{x})^2 - 2r(\mathbf{x}) \right]$$
$$+ (1 - \alpha)\mathbb{E}_{p_{\text{tr}}(\mathbf{x})} \left[ r(\mathbf{x})^2 \right] + C, \quad (22)$$

where $p_\alpha(\mathbf{x}) := \alpha p_{\text{te}}(\mathbf{x}) + (1-\alpha)p_{\text{tr}}(\mathbf{x})$ and $C$ is a constant term that does not depend on $r$. The empirical estimate of $J$ can be obtained as

$$\widehat{J}(r) := \frac{1}{N_{\text{te}}} \sum_{n=1}^{N_{\text{te}}} \left[ \alpha r(\mathbf{x}_{\text{te},n})^2 - 2r(\mathbf{x}_{\text{te},n}) \right]$$
$$+ \frac{1 - \alpha}{N_{\text{tr}}} \sum_{n=1}^{N_{\text{tr}}} r(\mathbf{x}_{\text{tr},n})^2 + C. \quad (23)$$

### 4.4. Training Procedure

To calculate $\widehat{\mathcal{R}}_\sigma^{\text{te}}$ in Eq. (20), the standard approach is to first estimate the importance weights and then use them to calculate the importance-weighted risk (Yamada et al., 2013; Sugiyama et al., 2012; Kanamori et al., 2009). However, in

this two-step approach, errors in the importance weight estimation propagate to the subsequent importance-weighted risk calculation, which degrades the performance of the learned classifiers (Fang et al., 2020; Zhang et al., 2020). To alleviate this, Fang et al. (2020; 2023) recently proposed a dynamic approach that iterates between the importance weight estimation and classifier learning for ordinary supervised learning.

The proposed method follows this dynamic approach. Specifically, we use the following neural networks for modeling the score function and importance weight,

$$s(\mathbf{x}) := u(h(\mathbf{x})), \ \ r(\mathbf{x}) := v(h(\mathbf{x})), \quad (24)$$

where $h : \mathcal{X} \to \mathbb{R}^K$, $u : \mathbb{R}^K \to \mathbb{R}$, and $v : \mathbb{R}^K \to \mathbb{R}$ are neural networks for feature extraction, score function, and importance weights, respectively. By sharing feature extractor $h$, we can effectively perform the importance weighting.

Algorithm 1 shows the training procedure of the proposed method with stochastic gradient descent methods. We first randomly sample PU data from the training distribution and unlabeled data in the test distribution (Lines 2–3). Then, we calculate the loss in Eq. (23) for the importance weight estimation (Line 4) and update parameters of $v$ with the gradient of the loss fixing feature extractor $h$ (Line 5). We fixed $h$ to avoid overfitting as in (Fang et al., 2020). By using the estimated importance weights, we calculate the loss in Eq. (20) for classifier learning (Line 6). We update parameters of both $u$ and $h$ by using the gradient of the loss (Line 7). In this step, we fixed the importance weights to avoid learning a meaningless model, $r(\mathbf{x}) = 0$ for all $\mathbf{x}$.

## 5. Experiments

### 5.1. Data

We used four real-world datasets in the main paper: MNIST (LeCun et al., 1998), FashionMNIST (Xiao et al., 2017), SVHN (Netzer et al., 2011), and CIFAR10 (Krizhevsky et al., 2009). These datasets have been commonly used in PU learning or distribution adaptation studies (Kumagai et al., 2024; Fang et al., 2020; Xie et al., 2024; Kiryo et al., 2017; Jiang et al., 2023). MNIST consists of hand-written images of 10 digits. Each image is represented by gray-scale with $28 \times 28$ pixels. FashionMNIST consists of images of 10 fashion categories where each image is represented by gray scale with $28 \times 28$ pixels. SVHN consists of $32 \times 32$ RGB images of 10 printed digits clipped from photographs of house number plates. We converted SVHN into gray-scale for simplicity. CIFAR10 consists of $32 \times 32$ RGB images of 10 animal and vehicle categories. In Appendix D.3, we also used real-world tabular datasets: epsilon and Hreadmission (Gardner et al., 2023).

To create the situation of the covariate shift, we imposed a

---

**Algorithm 1** Training procedure of the proposed method

**Require:** PU data in the training distribution and unlabeled data in the test distribution $X_{\mathrm{tr}}^{\mathrm{P}} \cup X_{\mathrm{tr}} \cup X_{\mathrm{te}}$, mini-batch size $M$, positive mini-batch size $P$, relative parameter $\alpha$, and weighting parameter $\beta$.

**Ensure:** Parameters of neural networks $h$, $u$, and $v$.

1: **repeat**
2:      Sample unlabeled data with size $M$ form $X_{\mathrm{tr}} \cup X_{\mathrm{te}}$
3:      Sample positive data with size $P$ form $X_{\mathrm{tr}}^{\mathrm{P}}$
     {Importance weight estimation}
4:      Calculate the loss in Eq. (23) on the sampled data
5:      Update parameters of $v$ with the gradient of the loss fixing feature extractor $h$
     {Classifier learning}
6:      Calculate the loss in Eq. (20) on the sampled data with current importance weights
7:      Update parameters of $u$ and $h$ with the gradient of the loss fixing the importance weights
8: **until** End condition is satisfied;

---

selection bias in the training and testing distribution following a previous study (Aminian et al., 2022). Specifically, for MNIST and SVHN, we used even digits as negative and odd digits as positive. For the training distribution, 90% of data were selected from the digits 0, 1, 2, and 3, and 10% of data were selected from the remaining digits. We reversed this ratio for the test distribution: 10% of data were selected from the digits 0, 1, 2, and 3, and 90% of data were selected from the remaining digits. For FashionMNIST, following the study (Xie et al., 2024; Kumagai et al., 2024), we used upper garments (0, 2, 3, 4, and 6) as negative and the others as positive, where numbers in parentheses represent class labels. Similar to MNIST and SVHN, for the training distribution, 90% of data had the class labels 0, 1, 2, and 5, and 10% of data had the remaining class labels. We reversed this ratio for the test distribution. For CIFAR10, we used the animal categories (2, 3, 4, 5, 6, and 7) as negative and the vehicles as positive. For the training distribution, 90% of data had the class labels 0, 1, 2, 3, and 4, and 10% of data had the remaining class labels. We reversed this ratio for the test distribution.

For training in each dataset, we used 50 positive and 3,000 unlabeled data in the training distribution and 3,000 unlabeled data in the test distribution. Additionally, we used 20 positive and 250 unlabeled data in the training distribution and 250 unlabeled data in the test distribution for validation. For unlabeled data in training and validation, we changed class-prior $\pi := \pi_{\mathrm{tr}} = \pi_{\mathrm{te}}$ within $\{0.01, 0.05, 0.1\}$. We used 3,000 data in the test distribution as test data for evaluation. Training, validation, and test datasets did not overlap. We conducted 10 experiments for each positive class-prior changing the random seeds and evaluated mean test AUC.

## 5.2. Comparison Methods

We compared the proposed method with eight methods: non-negative PU learning method with PU data in the training distribution (trPU) (Kiryo et al., 2017), non-negative PU learning method with positive data in the training and unlabeled data in the test distributions (tePU), AUC maximization method with PU data in the training distribution (trAUC) (Xie & Li, 2018; Xie et al., 2024), AUC maximization method with positive data in the training and unlabeled data in the test distributions (teAUC), AUC maximization method with PU data in the training distribution and unlabeled data in the test distribution (trteAUC), unsupervised domain adaptation method with AUC maximization (UDAUC), AUC maximization method for positive distribution shift (PAUC) (Kumagai et al., 2024), and covariate shift adaptation method for PU learning (CPU) (Sakai & Shimizu, 2019). All methods including the proposed method used neural networks for modeling classifiers.

trPU learns the neural network by minimizing the non-negative PU risk with PU data in the training distribution. This method does not consider the class-imbalance. trAUC learns the neural network by maximizing the AUC with PU data in the training distribution. trAUC is equivalent to the proposed method that minimizes $\widehat{\mathcal{R}}_{\sigma,1}^{\text{te}}(s)$ in Eq. (17) without importance weights (i.e., $w(\mathbf{x}) = 1$ for all $\mathbf{x}$). trPU and trAUC do not adapt to the test distribution. tePU and teAUC use unlabeled data in the test distribution instead of unlabeled data in the training distribution within trPU and trAUC, respectively. Especially, teAUC is equivalent to the proposed method that minimizes $\widehat{\mathcal{R}}_{\sigma,2}^{\text{te}}(s)$ in Eq. (19) without importance weights (i.e., $w(\mathbf{x}) = 1$ for all $\mathbf{x}$).

trteAUC, UDAUC, PAUC, and CPU use PU data in the training distribution and unlabeled data in the test distribution as in the proposed method. trteAUC uses a weighted sum of the losses of trAUC and teAUCt for training. Thus, trteAUC is equivalent to the proposed method that minimizes $\widehat{\mathcal{R}}_{\sigma}^{\text{te}}(s)$ in Eq. (20) without importance weights (i.e., $w(\mathbf{x}) = 1$ for all $\mathbf{x}$). UDAUC learns the invariant feature representation to mitigate the discrepancy of the training and test distribution by using CORAL loss, which is a widely used in domain adaptation studies (Sun et al., 2016; 2017), as in the previous method (Yang et al., 2023). Specifically, this method minimizes the weighted sum of the loss of trAUC and the CORAL loss for training. PAUC is a recently proposed AUC maximization method for positive distribution shift, where negative-conditional density does not change but the positive-conditional density can vary between the training and test phases. CPU is a PU learning method for covariate shift adaptation. We used the dynamic approach for the importance weighting in CPU. This method does not consider the class-imbalance. For trPU, tePU, and CPU, the absolute-value risk correction was used to improve the

performance as in previous studies (Lu et al., 2020; Hammoudeh & Lowd, 2020). trPU, tePU, PAUC, and CPU used information on class-prior of the training distribution $\pi_{\text{tr}}$ for training although the proposed method does not require it.

## 5.3. Settings

For all methods, the sigmoid loss was used instead of indicator functions as in the previous studies (Kiryo et al., 2017; Kumagai et al., 2024). The empirical risk with validation data was used to select hyperparameters and early-stopping to mitigate overfitting. All methods were implemented using Pytorch (Paszke et al., 2017), and all experiments were conducted on a Linux server with an Intel Xeon CPU and A100 GPU. The details of neural network architectures and hyperparameters are described in Appendixes B and C.

## 5.4. Results

Table 1 shows the average test AUCs of each method on the four class-imbalanced datasets. Full results including the standard deviations are described in Appendix D.6. The proposed method performed the best or comparably to it in all cases. trPU and tePU did not work well since they cannot deal with the class-imbalance. trAUC and teAUC performed better than trPU and tePU, which indicates that the AUC maximization is appropriate for the class-imbalanced data. PAUC performed poorly because the shift assumption used in PAUC (i.e., positive distribution shift) is different to the covariate shift and it struggles with small $\pi_{\text{te}}$ due to its reliance on extracting positive data from unlabeled test data as described in the paper (Kumagai et al., 2024). Although CPU is a method for covariate shift, it did not work well because it does not consider the class-imbalance. Although UDAUC that learns invariant feature representations to mitigate the distribution gap performed better than other distribution shift methods (PAUC and CPU), it performed worse than the proposed method. This result suggests that the approach of learning invariant features was often sub-optimal for the covariate shift problem. The proposed method performed better than trteAUC, trAUC, and teAUC, which can be regarded as the special cases of the proposed method (i.e., non-importance weighting versions of our method) as described in Section 5.2. This result indicates that the importance weighting with AUC maximization is useful for the class-imbalanced data under the covariate shift. As for the results of each class-prior $\pi$, the performance of AUC-based methods (the proposed method, trAUC, teAUC, and trteAUC) tended to improve as the value of $\pi$ decreased. This is because they essentially use loss functions of the form, $\mathbb{E}_{\mathbf{x}^{\text{p}} \sim p^{\text{p}}(\mathbf{x})} \mathbb{E}_{\mathbf{x} \sim p(\mathbf{x})} [f(\mathbf{x}^{\text{p}}, \mathbf{x})]$, where $p^{\text{p}}(\mathbf{x})$ and $p(\mathbf{x})$ are positive and marginal densities. When $\pi$ is small, $p(\mathbf{x})$ can be regarded as negative density $p^{\text{n}}(\mathbf{x})$. In this case, the above loss function becomes the original AUC risk. Thus, these methods work well with small $\pi$. Since PAUC has a

*Table 1.* Average test AUCs with different positive class-prior $\pi := \pi_{\text{tr}} = \pi_{\text{te}}$. Values in bold are not statistically different at the 5% level from the best performing method in each row according to a paired t-test. '# best' row represents the number of results of each method that are the best or comparable to it.

| Data | $\pi$ | Ours | trPU | tePU | trAUC | teAUC | trteAUC | PAUC | UDAUC | CPU |
|------|-------|------|------|------|-------|-------|---------|------|-------|-----|
| MNIST | 0.01 | **0.817** | 0.677 | 0.654 | 0.795 | **0.813** | 0.795 | 0.402 | 0.794 | 0.667 |
| | 0.05 | **0.812** | 0.750 | 0.695 | **0.793** | 0.806 | 0.806 | 0.473 | **0.795** | 0.734 |
| | 0.1 | **0.783** | **0.764** | 0.691 | **0.774** | **0.779** | **0.779** | 0.504 | **0.774** | 0.748 |
| Fashion | 0.01 | **0.925** | **0.912** | **0.906** | 0.920 | 0.912 | 0.920 | 0.634 | **0.926** | **0.920** |
| MNIST | 0.05 | **0.907** | **0.902** | **0.890** | **0.906** | 0.871 | 0.871 | 0.730 | **0.906** | **0.893** |
| | 0.1 | **0.890** | 0.834 | 0.790 | **0.899** | 0.839 | 0.839 | 0.718 | **0.899** | 0.829 |
| SVHN | 0.01 | **0.715** | 0.504 | 0.504 | 0.548 | **0.724** | **0.725** | 0.236 | 0.549 | 0.504 |
| | 0.05 | **0.686** | 0.504 | 0.504 | 0.559 | **0.698** | **0.698** | 0.246 | 0.559 | 0.504 |
| | 0.1 | **0.666** | 0.504 | 0.504 | 0.533 | **0.677** | **0.677** | 0.266 | 0.540 | 0.504 |
| CIFAR10 | 0.01 | **0.896** | 0.456 | 0.475 | 0.874 | 0.874 | 0.874 | 0.364 | 0.873 | 0.508 |
| | 0.05 | **0.893** | 0.456 | 0.482 | **0.878** | **0.884** | **0.884** | 0.475 | **0.871** | 0.534 |
| | 0.1 | **0.870** | 0.469 | 0.587 | **0.871** | **0.874** | **0.874** | 0.560 | **0.869** | 0.542 |
| # best | | 12 | 3 | 2 | 6 | 7 | 6 | 0 | 7 | 2 |

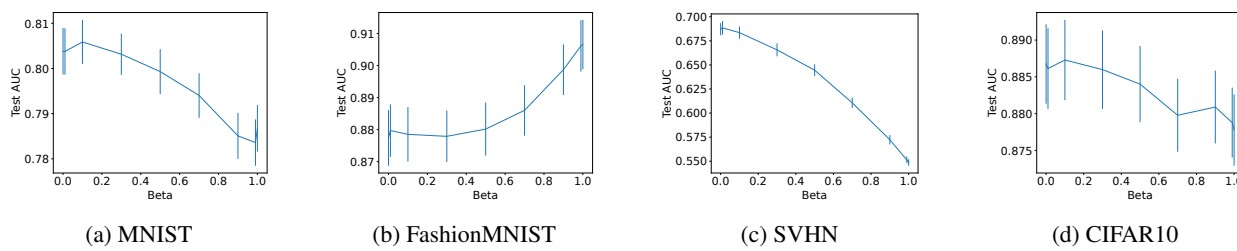

(a) MNIST     (b) FashionMNIST     (c) SVHN     (d) CIFAR10

*Figure 1.* The average test AUCs with the standard errors of the proposed method when changing weighting hyperparameter $\beta$.

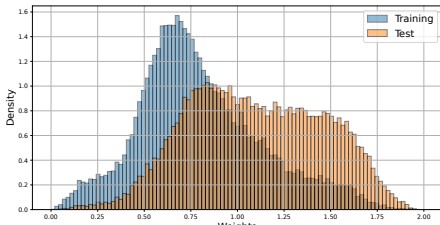

*Figure 2.* Importance weight distribution of our method with FashionMNIST when $\pi = 0.1$. 'Training' (blue) and 'Test' (orange) represent data in the training and test distributions, respectively.

*Table 2.* Comparison with the two-step importance weighting of the proposed method (Ours w/ 2step). Average test AUCs over different positive class-priors on each dataset. FMNIST is an abbreviation of FashionMNIST.

| | MNIST | FMNIST | SVHN | CIFAR10 |
|--|-------|--------|------|---------|
| Ours | **0.817** | **0.907** | **0.690** | **0.886** |
| Ours w/ 2step | **0.820** | **0.911** | 0.608 | 0.876 |

CIFAR10, smaller values of $\beta$ tended to yield better results, while for FashionMNIST, larger values of $\beta$ led to better results. The proposed method was able to select good $\beta$ by using validation data. These results show the effectiveness of using the weighted sum of two AUC risk estimators.

Figure 2 visualizes the distribution of the importance weights estimated by the proposed method with FashionMNIST. The data in the test distribution tended to have larger importance weights than the data in the training distribution. This result shows that the proposed method can estimate importance weights as expected.

Table 2 compares the proposed method and Ours w/ 2step, which is the proposed method using conventional two-step importance weighting. As expected, the proposed method, which uses the dynamic approach, tended to perform better

different form of loss, its trend was different.

Figure 1 shows the average test AUCs with the standard errors of the proposed method when changing $\beta$, which controls the effect of two estimators of the AUC risk on the test distribution. When $\beta = 1$, our loss function becomes the AUC risk estimator with PU data in the training distribution (Eq. (17)). When $\beta = 0$, our loss function becomes the AUC risk estimator with positive data in the training and unlabeled data in the test distribution (Eq. (19)). The trends of the results varied across datasets: for MNIST, SVHN, and

*Table 3.* Results under class-prior shift: average test AUCs with different positive test class-prior $\pi_{\text{te}}$ with postive training class-prior $\pi_{\text{tr}} = 0.01$. Values in bold are not statistically different at the 5% level from the best performing method in each row according to a paired t-test. '# best' row represents the number of results of each method that are the best or comparable to it.

| Data | $\pi_{\text{te}}$ | Ours | trPU | tePU | trAUC | teAUC | trteAUC | PAUC | UDAUC | CPU |
|---|---|---|---|---|---|---|---|---|---|---|
| MNIST | 0.01 | **0.938** | 0.729 | 0.729 | **0.938** | **0.938** | **0.938** | 0.562 | **0.938** | 0.729 |
| | 0.05 | **0.929** | 0.734 | 0.717 | **0.930** | 0.924 | **0.930** | 0.762 | **0.930** | 0.733 |
| | 0.1 | **0.930** | 0.736 | 0.709 | **0.932** | 0.921 | **0.932** | 0.866 | **0.932** | 0.731 |
| Fashion | 0.01 | **0.983** | 0.899 | 0.895 | **0.982** | **0.984** | **0.984** | 0.818 | **0.982** | 0.906 |
| MNIST | 0.05 | **0.983** | 0.900 | 0.770 | **0.983** | 0.980 | **0.983** | 0.929 | **0.983** | 0.897 |
| | 0.1 | **0.988** | 0.901 | 0.743 | **0.988** | 0.983 | **0.988** | 0.970 | **0.988** | 0.895 |
| SVHN | 0.01 | **0.644** | 0.504 | 0.504 | **0.642** | 0.632 | **0.655** | 0.499 | **0.648** | 0.504 |
| | 0.05 | **0.648** | 0.504 | 0.504 | **0.625** | 0.600 | **0.635** | 0.505 | **0.624** | 0.504 |
| | 0.1 | **0.639** | 0.504 | 0.504 | **0.641** | 0.619 | **0.648** | 0.507 | **0.645** | 0.504 |
| CIFAR10 | 0.01 | **0.884** | 0.432 | 0.427 | 0.867 | 0.872 | 0.872 | 0.519 | 0.866 | 0.481 |
| | 0.05 | **0.890** | 0.433 | 0.424 | 0.878 | 0.872 | 0.878 | 0.680 | 0.878 | 0.480 |
| | 0.1 | **0.891** | 0.432 | 0.424 | **0.885** | 0.867 | **0.886** | 0.785 | **0.885** | 0.482 |
| # best | | 12 | 0 | 0 | 10 | 3 | 10 | 0 | 10 | 0 |

than Ours w/ 2step. Note that Ours w/ 2step is also our proposal since there are no existing importance weighting methods for AUC maximization.

Although the proposed method assumes the covariate shift, other forms of distribution shifts may arise in real-world scenarios. Therefore, we additionally evaluated the proposed method under a class-prior shift, in which the class-prior changes, but the class-conditional density remains the same (Lu et al., 2022). Table 3 shows the results. The proposed method empirically worked well. This would be because the proposed method does not depend on class-priors and thus is relatively robust against the class-prior shift.

Additionally, scenarios without any distribution shift may also arise in practice. Table 4 shows the results without distribution shifts (i.e., $p_{\text{tr}}(\mathbf{x}, y) = p_{\text{te}}(\mathbf{x}, y)$). Since there were no shifts, we compared trPU and trAUC, which do not consider shifts. The proposed method and trAUC showed comparable results, indicating that the proposed method robustly works well when no shift exists.

## 6. Conclusion

In this paper, we proposed a method for maximizing the AUC under a covariate shift. To construct the method, we theoretically derived two estimators of the AUC risk on the test distribution: the first is calculated from importance-weighted PU data in the training distribution, and the second is calculated from importance-weighted positive data in the training distribution and unlabeled data in the test distribution. Our loss function for classifier learning is a weighted sum of these two AUC risk estimators. The experiments show that our method outperformed various existing PU learning and distribution adaptation methods.

*Table 4.* Results without distribution shift: average test AUCs over different positive class-prior $\pi := \pi_{\text{tr}} = \pi_{\text{te}}$ within $\{0.01, 0.05, 0.1\}$.

| Data | Ours | trPU | trAUC |
|---|---|---|---|
| MNIST | 0.927 | 0.814 | 0.927 |
| FashionMNIST | 0.982 | 0.916 | 0.982 |
| SVHN | 0.625 | 0.504 | 0.630 |
| CIFAR10 | 0.885 | 0.436 | 0.875 |

## 7. Limitations

The proposed method assumes a specific type of distribution shift: the covariate shift. Although we found that the proposed method works well under the covariate and class-prior shifts in our experiments and we can often expect the validity of the covariate shift assumption to a certain extent in each real-world application, the proposed method is not guaranteed to work well if other distribution shifts occur. Therefore, a method should preferably be developed that can estimate the type of distribution shifts from data.

## Impact Statement

Although our method performed well, misclassification is possible in practice. In particular, misclassification can lead to serious incidents in cases such as cyber security and medical care, which are typical examples of imbalanced data. Thus, our method should be used as a support tool for humans to make a final decision.

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

## A. Derivation of Eq. (15)

**Lemma A.1.** $\mathbb{E}_{\mathbf{x}^{\mathrm{P}} \sim p_{\mathrm{tr}}^{\mathrm{P}}(\mathbf{x})} \mathbb{E}_{\bar{\mathbf{x}}^{\mathrm{P}} \sim p_{\mathrm{tr}}^{\mathrm{P}}(\mathbf{x})} \left[ w(\mathbf{x}^{\mathrm{P}}) w(\bar{\mathbf{x}}^{\mathrm{P}}) f(\mathbf{x}^{\mathrm{P}}, \bar{\mathbf{x}}^{\mathrm{P}}) \right] = \frac{1}{2} \mathbb{E}_{\mathbf{x}^{\mathrm{P}} \sim p_{\mathrm{tr}}^{\mathrm{P}}(\mathbf{x})} \mathbb{E}_{\bar{\mathbf{x}}^{\mathrm{P}} \sim p_{\mathrm{tr}}^{\mathrm{P}}(\mathbf{x})} \left[ w(\mathbf{x}^{\mathrm{P}}) w(\bar{\mathbf{x}}^{\mathrm{P}}) \right] = C$, *where $C$ is a constant that does not depend on $s$.*

*Proof.* Since $f(\mathbf{x}^{\mathrm{P}}, \bar{\mathbf{x}}^{\mathrm{P}}) = \sigma(-s(\mathbf{x}^{\mathrm{P}}) + s(\bar{\mathbf{x}}^{\mathrm{P}}))$ and $\sigma(z) + \sigma(-z) = 1$ for all $z \in \mathbb{R}$,

$$
\begin{aligned}
\mathbb{E}_{\mathbf{x}^{\mathrm{P}} \sim p_{\mathrm{tr}}^{\mathrm{P}}(\mathbf{x})} \mathbb{E}_{\bar{\mathbf{x}}^{\mathrm{P}} \sim p_{\mathrm{tr}}^{\mathrm{P}}(\mathbf{x})} \left[ w(\mathbf{x}^{\mathrm{P}}) w(\bar{\mathbf{x}}^{\mathrm{P}}) f(\mathbf{x}^{\mathrm{P}}, \bar{\mathbf{x}}^{\mathrm{P}}) \right] = & \mathbb{E}_{\mathbf{x}^{\mathrm{P}} \sim p_{\mathrm{tr}}^{\mathrm{P}}(\mathbf{x})} \mathbb{E}_{\bar{\mathbf{x}}^{\mathrm{P}} \sim p_{\mathrm{tr}}^{\mathrm{P}}(\mathbf{x})} \left[ w(\mathbf{x}^{\mathrm{P}}) w(\bar{\mathbf{x}}^{\mathrm{P}}) \left[ 1 - f(\bar{\mathbf{x}}^{\mathrm{P}}, \mathbf{x}^{\mathrm{P}}) \right] \right] \\
= & \mathbb{E}_{\mathbf{x}^{\mathrm{P}} \sim p_{\mathrm{tr}}^{\mathrm{P}}(\mathbf{x})} \mathbb{E}_{\bar{\mathbf{x}}^{\mathrm{P}} \sim p_{\mathrm{tr}}^{\mathrm{P}}(\mathbf{x})} \left[ w(\mathbf{x}^{\mathrm{P}}) w(\bar{\mathbf{x}}^{\mathrm{P}}) \right] \\
& - \mathbb{E}_{\mathbf{x}^{\mathrm{P}} \sim p_{\mathrm{tr}}^{\mathrm{P}}(\mathbf{x})} \mathbb{E}_{\bar{\mathbf{x}}^{\mathrm{P}} \sim p_{\mathrm{tr}}^{\mathrm{P}}(\mathbf{x})} \left[ w(\mathbf{x}^{\mathrm{P}}) w(\bar{\mathbf{x}}^{\mathrm{P}}) f(\bar{\mathbf{x}}^{\mathrm{P}}, \mathbf{x}^{\mathrm{P}}) \right].
\end{aligned} \tag{25}
$$

It is clear that the lemma follows from this equation. $\qquad\square$

## B. Neural Network Architectures

For MNIST, FashionMNIST, and SVHN, a three-layered feed-forward neural network with ReLU activation was used for the feature extractor $h$. The number of hidden and output nodes was 32. For CIFAR10, a convolutional neural network, which consisted of two convolutional blocks followed by a two-layered feed-forward neural network, was used for the feature extractor $h$. The first (second) convolutional block comprised a 6 (16) filter $5 \times 5$ convolution, the ReLU activation, and a $2 \times 2$ max-pooling layer. The numbers of the hidden and output nodes were 120 and 84, respectively. One-layered and two-layered feed-forward neural networks were used for $u$ and $v$, respectively. For the output activation of $v$, we used $\frac{1}{\alpha}\sigma(\cdot)$, where $\sigma$ is a sigmoid function, to match the value range of the relative density-ratio (importance weights) when $\alpha > 0$. When $\alpha = 0$ (i.e., using ordinary density-ratio), we used the softplus function for output activation. For all comparison methods, the same neural network architecture (i.e., $u(h(\mathbf{x}))$) was used for the classifier (score function). For CPU, the architecture of $v$ was also the same as that of the proposed method.

## C. Hyperparameters

For all methods, the empirical risk with validation data was used to select hyperparameters and early-stopping to mitigate overfitting. Specifically, for the proposed method, the weighted risk in Eq. (20) was used. For the proposed method and CPU, relative parameter $\alpha$ was set to 0.5 for all datasets. For the proposed method and trteAUC, weighting parameter $\beta$ was selected from $\{0.0, 0.01, 0.1, 0.3, 0.5, 0.7, 0.9, 0.99, 1.0\}$. For UDAUC, the CORAL loss was applied to $h(\mathbf{x})$. The weighting parameter of the CORAL loss was chosen from $\{1, 10^{-1}, 10^{-2}, 10^{-3}\}$. The mini-batch size $M$ was set to 512 and the positive mini-batch size $P$ was set to 50. For all methods, we used the Adam optimizer (Kingma & Ba, 2014). We set the learning rate to $10^{-4}$. The maximum number of epochs was 200. All methods were implemented using Pytorch (Paszke et al., 2017), and all experiments were conducted on a Linux server with an Intel Xeon CPU and A100 GPU.

## D. Additional Experimental Results

### D.1. Results with Different Numbers of Labeled Positive Data in the Training Distribution

Figure 3 shows the average test AUCs with the standard error of the proposed method with different numbers of labeled positive data in the training distribution $N_{\mathrm{tr}}^{\mathrm{P}}$. As expected, the performance of the proposed method improved as $N_{\mathrm{tr}}^{\mathrm{P}}$ increased.

### D.2. Results with Different Relative Parameters $\alpha$

Figure 4 shows the average test AUCs with the standard error of the proposed method with different values of relative parameter $\alpha$. Although the tendency of the results varied across datasets, the proposed method with $\alpha = 0.5$ worked well for all datasets.

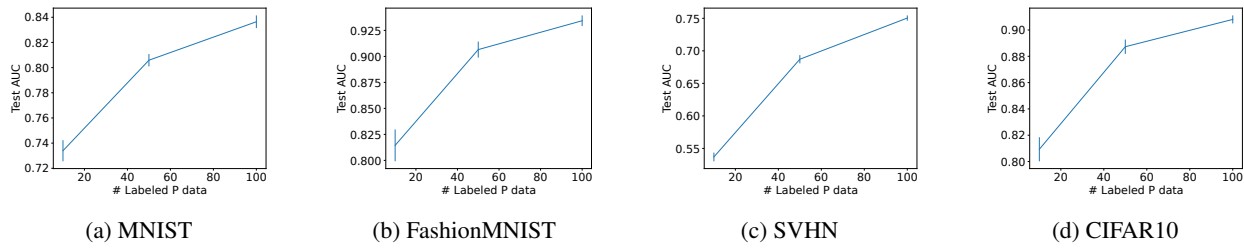

*Figure 3.* The average test AUCs with the standard errors of the proposed method when changing the number of labeled positive data in the training distribution $N_{\mathrm{tr}}^{\mathrm{P}}$.

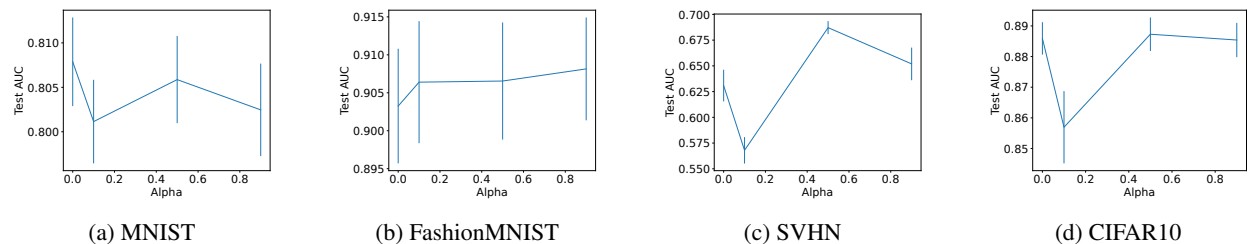

*Figure 4.* The average test AUCs with the standard errors of the proposed method when changing relative hyperparameter $\alpha$.

*Table 5.* Average test AUCs over different positive class-prior $\pi := \pi_{\mathrm{tr}} = \pi_{\mathrm{te}}$ within $\{0.01, 0.05, 0.1\}$ on tabular datasets. Values in bold are not statistically different at the $5\%$ level from the best performing method in each row according to a paired t-test.

| Data | Ours | trPU | tePU | trAUC | teAUC | trteAUC | PAUC | UDAUC | CPU |
|------|------|------|------|-------|-------|---------|------|-------|-----|
| epsilon | **0.530** | 0.479 | 0.480 | **0.529** | **0.530** | **0.530** | 0.514 | **0.528** | 0.490 |
| Hreadmission | **0.694** | 0.541 | 0.561 | **0.700** | 0.553 | 0.553 | 0.477 | **0.699** | 0.551 |

## D.3. Results with Tabular Data with Distribution Shifts

We evaluated the proposed method with two tabular datasets: epsilon [2] and Hreadmission (Gardner et al., 2023). In epsilon, the feature vector size is 2,000. To create the covariate shift, we followed the procedure used in a previous study (Sakai & Shimizu, 2019). Specifically, we first calculated the Euclidean distance between instance $\mathbf{x}_n$ and the mean vector of all data $\bar{\mathbf{x}}$, $\mathbf{c}_n := \|\mathbf{x} - \bar{\mathbf{x}}\|$. Then, we found the median $\mathbf{c}_{\mathrm{med}}$ from all $\{\mathbf{c}_n\}$. We split all $\{\mathbf{c}_n\}$ into the first set whose elements were smaller than $\mathbf{c}_{\mathrm{med}}$ and the second set whose elements were larger than $\mathbf{c}_{\mathrm{med}}$. With probability 0.9 and 0.1, instances whose indices were in the first set were selected as data in the training and test distributions, respectively. In contrast, instances whose indices were in the second set were selected as data in the training and test distributions with probability 0.1 and 0.9, respectively. In Hreadmission, the task is to predict the 30-day readmission of diabetic hospital patients. Each patient is represented by a 183-dimensional feature vector. This dataset has a distribution shift where the training and test distributions were created by "admission source". The experimental settings such as the number of data were the same as those in the main paper. Table 5 shows the average test AUCs on these two tabular datasets. The proposed method performed the best or comparably to it in all cases.

## D.4. Computation Cost

We investigated the training time of the proposed method on MNIST with $\pi = 0.1$. We used a Linux server with a 2.20Hz Central Processing Unit. For comparison, we also evaluated the methods that use PU data in the training distribution and unlabeled data in the test distribution as in the proposed method. Table 6 shows the results. Since the proposed method and CPU learned both importance weights and classifiers, they had slightly longer training times than the other methods. However, the differences were not significant. This result indicates that the proposed method is practical in terms of computation costs.

---

[2] https://www.csie.ntu.edu.tw/ cjlin/libsvmtools/datasets/

*Table 6.* Training time [s] of the proposed method on MNIST.

| Ours | trteAUC | PAUC | UDAUC | CPU |
|------|---------|------|-------|-----|
| 90.2 | 86.83 | 87.37 | 87.93 | 89.49 |

*Table 7.* Performance comparison across different learning rates.

| Method | Data | $10^{-6}$ | $10^{-5}$ | $10^{-4}$ | $10^{-3}$ | $10^{-2}$ | $10^{-1}$ |
|--------|------|-----------|-----------|-----------|-----------|-----------|-----------|
| Ours | MNIST | 0.709 | 0.760 | 0.804 | 0.806 | 0.796 | 0.738 |
| Ours | FahionFMNIST | 0.854 | 0.914 | 0.907 | 0.873 | 0.874 | 0.789 |
| Ours | SVHN | 0.507 | 0.532 | 0.689 | 0.682 | 0.669 | 0.564 |
| Ours | CIFAR10 | 0.810 | 0.897 | 0.886 | 0.884 | 0.789 | 0.644 |
| trAUC | MNIST | 0.693 | 0.756 | 0.787 | 0.785 | 0.779 | 0.784 |
| trAUC | FashionMNIST | 0.846 | 0.916 | 0.909 | 0.895 | 0.891 | 0.876 |
| trAUC | SVHN | 0.500 | 0.503 | 0.547 | 0.553 | 0.537 | 0.533 |
| trAUC | CIFAR10 | 0.761 | 0.879 | 0.874 | 0.870 | 0.865 | 0.738 |

*Table 8.* Average test AUCs and their standard deviations with different positive class-prior $\pi := \pi_{\mathrm{tr}} = \pi_{\mathrm{te}}$. Values in bold are not statistically different at the $5\%$ level from the best performing method in each row according to a paired t-test.

| Data | $\pi$ | Ours | trPU | tePU | trAUC | teAUC | trteAUC | PAUC | UDAUC | CPU |
|------|-------|------|------|------|-------|-------|---------|------|-------|-----|
| MNIST | 0.01 | **0.817(0.021)** | 0.677(0.020) | 0.654(0.009) | 0.795(0.226) | **0.813(0.022)** | 0.795(0.023) | 0.402(0.313) | 0.794(0.023) | 0.667(0.008) |
| | 0.05 | **0.812(0.024)** | 0.750(0.047) | 0.695(0.022) | **0.793(0.028)** | 0.806(0.027) | 0.806(0.027) | 0.473(0.043) | **0.795(0.029)** | 0.734(0.038) |
| | 0.1 | **0.783(0.027)** | **0.764(0.026)** | 0.691(0.038) | **0.774(0.022)** | **0.779(0.026)** | **0.779(0.026)** | 0.504(0.043) | **0.774(0.022)** | 0.748(0.037) |
| Fashion | 0.01 | **0.925(0.028)** | **0.912(0.032)** | **0.906(0.029)** | 0.920(0.027) | 0.912(0.021) | 0.920(0.027) | 0.634(0.038) | **0.926(0.026)** | **0.920(0.029)** |
| MNIST | 0.05 | **0.907(0.040)** | **0.902(0.067)** | **0.890(0.068)** | **0.906(0.036)** | 0.871(0.033) | 0.871(0.033) | 0.730(0.031) | **0.906(0.036)** | **0.893(0.063)** |
| | 0.1 | **0.890(0.048)** | 0.834(0.053) | 0.790(0.046) | **0.899(0.047)** | 0.839(0.045) | 0.839(0.045) | 0.718(0.035) | **0.899(0.047)** | 0.829(0.058) |
| SVHN | 0.01 | **0.715(0.016)** | 0.504(0.008) | 0.504(0.008) | 0.548(0.022) | **0.724(0.024)** | **0.725(0.027)** | 0.236(0.006) | 0.549(0.020) | 0.504(0.008) |
| | 0.05 | **0.686(0.024)** | 0.504(0.009) | 0.504(0.008) | 0.559(0.016) | **0.698(0.014)** | **0.698(0.014)** | 0.246(0.007) | 0.559(0.013) | 0.504(0.008) |
| | 0.1 | **0.666(0.037)** | 0.504(0.008) | 0.504(0.008) | 0.533(0.024) | **0.677(0.040)** | **0.677(0.040)** | 0.266(0.009) | 0.540(0.024) | 0.504(0.008) |
| CIFAR10 | 0.01 | **0.896(0.016)** | 0.456(0.070) | 0.475(0.095) | 0.874(0.013) | 0.874(0.025) | 0.874(0.025) | 0.364(0.039) | 0.873(0.013) | 0.508(0.075) |
| | 0.05 | **0.893(0.025)** | 0.456(0.064) | 0.482(0.114) | **0.878(0.020)** | **0.884(0.026)** | **0.884(0.026)** | 0.475(0.046) | **0.871(0.021)** | 0.534(0.115) |
| | 0.1 | **0.870(0.038)** | 0.469(0.071) | 0.587(0.174) | **0.871(0.024)** | **0.874(0.025)** | **0.874(0.025)** | 0.560(0.033) | **0.869(0.027)** | 0.542(0.120) |

## D.5. Results with Different Learning Rates

We investigated the performance obtained by varying the learning rates of the Adam optimizer. Table 7 shows the average test AUCs over different class-prior within $\{0.01, 0.05, 0.1\}$ of the proposed method and trAUC, which is the most basic baseline. As observed, the value $10^{-4}$ used in our experiments consistently shows good performance across all datasets.

## D.6. Full Results with Standard Deviations

Table 8 shows the average test AUCs and their standard deviations for each method. As class-prior $\pi$ decreased, the standard deviations of the proposed method also decreased. This is because when $\pi$ is small, unlabeled data can be approximately regarded as negative data; ranking function $f$ in Eqs. (17) and (19) can stably rank data well such that the scores of positive data are higher than those of negative data. Thus, its training becomes easier and more stable.

## D.7. Results with Larger Data

We performed the additional experiments on larger datasets. In this experiment, for each dataset, we used 100 positive and 9,000 unlabeled data in the training distribution and 9,000 unlabeled data in the test distribution for training. In addition, we included the recent PU learning method (PURA) (Jiang et al., 2023) for comparison. Since it is designed for ordinary PU learning, it used PU data in the training distribution. Margin $\rho$ was selected from $\{0.1, 1, 10\}$ by validation data. The results are described in Table 9. The proposed method outperformed the other methods. Since PURA does not consider distribution shift, it did not work well.

*Table 9.* Results in the large-data regime: average test AUCs with different positive class-prior $\pi := \pi_{\mathrm{tr}} = \pi_{\mathrm{te}}$. Values in bold are not statistically different at the 5% level from the best performing method in each row according to a paired t-test. '# best' row represents the number of results of each method that are the best or comparable to it.

| Data | $\pi$ | Ours | trPU | tePU | trAUC | teAUC | trteAUC | PAUC | UDAUC | CPU | PURA |
|---|---|---|---|---|---|---|---|---|---|---|---|
| MNIST | 0.01 | **0.849** | 0.719 | 0.690 | 0.823 | **0.842** | 0.823 | 0.417 | 0.823 | 0.704 | 0.808 |
| | 0.05 | **0.829** | 0.766 | 0.738 | 0.813 | 0.814 | 0.813 | 0.507 | 0.813 | 0.774 | 0.775 |
| | 0.1 | **0.810** | 0.779 | 0.736 | **0.806** | **0.805** | **0.805** | 0.524 | **0.806** | 0.757 | 0.776 |
| Fashion | 0.01 | **0.942** | **0.928** | 0.922 | 0.942 | 0.925 | 0.942 | 0.668 | **0.945** | **0.920** | 0.924 |
| MNIST | 0.05 | **0.935** | 0.864 | 0.750 | **0.945** | 0.912 | 0.912 | 0.728 | **0.946** | 0.854 | **0.952** |
| | 0.1 | **0.913** | 0.897 | 0.820 | 0.925 | 0.854 | 0.854 | 0.701 | 0.926 | 0.898 | **0.946** |
| SVHN | 0.01 | 0.763 | 0.504 | 0.504 | 0.628 | **0.780** | **0.780** | 0.252 | 0.628 | 0.504 | 0.503 |
| | 0.05 | **0.757** | 0.504 | 0.504 | 0.599 | **0.760** | **0.765** | 0.272 | 0.605 | 0.504 | 0.506 |
| | 0.1 | **0.750** | 0.504 | 0.504 | 0.587 | **0.749** | **0.751** | 0.304 | 0.586 | 0.504 | 0.503 |
| CIFAR10 | 0.01 | **0.920** | 0.450 | 0.476 | 0.915 | **0.922** | **0.922** | 0.343 | 0.914 | 0.509 | 0.854 |
| | 0.05 | **0.910** | 0.464 | 0.520 | **0.904** | **0.910** | **0.910** | 0.487 | **0.906** | 0.521 | 0.849 |
| | 0.1 | **0.900** | 0.552 | 0.654 | **0.885** | 0.877 | 0.879 | 0.547 | **0.900** | 0.579 | 0.864 |
| # best | | 11 | 1 | 0 | 4 | 8 | 7 | 0 | 5 | 1 | 2 |

*Table 10.* Results with the Food101 dataset: average test AUCs with different positive class-prior $\pi := \pi_{\mathrm{tr}} = \pi_{\mathrm{te}}$.

| $\pi$ | Ours | trPU | tePU | trAUC | teAUC | trteAUC | PAUC | UDAUC | CPU |
|---|---|---|---|---|---|---|---|---|---|
| 0.01 | 0.590 | 0.470 | 0.563 | 0.585 | 0.601 | 0.585 | 0.385 | 0.577 | 0.509 |
| 0.1 | 0.585 | 0.488 | 0.551 | 0.578 | 0.554 | 0.578 | 0.441 | 0.590 | 0.506 |
| avg | 0.588 | 0.479 | 0.557 | 0.582 | 0.578 | 0.582 | 0.413 | 0.584 | 0.508 |

## D.8. Results with the Food101 dataset

We additionally evaluated the proposed method with the Food101 dataset (Bossard et al., 2014) and the ResNet-18 model (He et al., 2016), which allows us to evaluate our method with a larger dataset and model. The Food101 dataset consists of image data from 101 food categories and is widely used for image classification tasks. The maximum length of each image is 512 pixels. We resized each image to $224 \times 224$ pixels. To create a binary classification problem, we divided the original 101 categories into sweets-related (positive) and main dish-related (negative) classes. Then, following the procedure described in Section 5.1, we split the original categories within each positive/negative into two groups, assigning the first half with smaller class indices to the first group and the remaining to the second group. We then created the covariate shift by using the group ratio of 9:1 for the training and 1:9 for the testing. We used 2,500 (200) positive and 25,000 (2,000) unlabeled training data and 25,000 (2,000) unlabeled test data for training (validation). We used 3,000 test data for evaluation. As for ResNet-18, we did not use pre-trained weights to purely investigate the performance with the given data. Table 10 shows the results. The proposed method performed slightly better than these methods on average.

