# OpenReview forum: "Positive-unlabeled AUC Maximization under Covariate Shift"
_ICML.cc/2025/Conference — ICML 2025 poster_

### Official Review · Reviewer_9PVZ · 2025-03-09

**Overall Recommendation:** 3

**Summary:**

In this paper, the authors focus on addressing the problem of AUC maximization under distribution shifts between training and testing data, specifically under the scenario of covariate shift, where the input distributions of the training and test data differ, but the conditional distribution of the class labels remains unchanged. To tackle this issue, the authors theoretically derive two estimators of the AUC risk, and combine positive instances with unlabeled data for training, without the need for negative examples or class-prior information. Extensive experiments on benchmark datasets validate the effectiveness of the proposed method.

**Claims And Evidence:**

The claims presented in the paper are supported by clear theoretical derivations and experimental validation.

**Essential References Not Discussed:**

N/A

**Experimental Designs Or Analyses:**

Overall, the experiments are comprehensive, but I still have a few concerns: these include the absence of experiments under noisy conditions and concerns regarding some outliers in the experimental results. The specific details will be provided in the Questions.

**Methods And Evaluation Criteria:**

The proposed method fills the gap in current AUC maximization techniques under covariate shift.

**Other Comments Or Suggestions:**

(1)	Has the issue of noise been considered? When the unlabeled data contains significant noise or redundant information, what impact would this have on the performance of the proposed AUC maximization method?
(2)	The paper explores a new scenario of "covariate shift" and argues that this scenario more closely approximates real-world situations. However, in real-world applications, such inconsistencies might also manifest as changes in class distributions (positive distribution shift) or shifts in class priors. In these inconsistent scenarios, can the proposed method still effectively perform AUC maximization? Does the paper address these issues and provide corresponding designs? Is the proposed method solely applicable to covariate shift?

**Other Strengths And Weaknesses:**

Strengths:
(1)	The method proposed in the paper is innovative and addresses a key issue regarding covariate shift in AUC maximization, an area that has not yet been explored in the literature.
(2)	The two estimators of the test AUC risk under covariate shift proposed in the paper are effective. This is demonstrated both theoretically and experimentally.
(3)	The paper is well-written and easy to understand.
Weaknesses:
(1)	The paper lacks a clear explanation of the motivation for deriving the two estimators of the AUC risk on the test distribution under covariate shift.
(2)	The proposed estimators are not unbiased. But there is no analysis of their error bounds.
(3)	The structure of the paper is essentially identical to that of Kumagai et al. (2024).

**Questions For Authors:**

(1)	The design in the paper relies on the sigmoid loss function. Would other loss functions be compatible with this solution approach?
(2)	Why do the PAUC results in Table 1 perform particularly poorly, even being the worst in most cases? They are troublingly bad.
(3)	The results on the SVHN dataset are consistently anomalous. For example, in Table 1, the UDAUC method exhibits unexpectedly poor results on the SVHN dataset compared to the other three datasets. In Table 2, the experimental results on SVHN show an anomalous reverse pattern compared to the other datasets.
(4)	Fig.2 shows the importance weight distribution on the FashionMNIST dataset when the positive class prior is 0.1. Does this result suggest that the method might overly rely on the test data during training? Could this lead to overfitting the test data? Furthermore, how do the distributions behave when the priors are set to 0.01/0.05? Would they show more pronounced features?

**Relation To Broader Scientific Literature:**

The contribution of this paper lies in proposing a new AUC optimization method that addresses covariate shift in weakly supervised learning, building on Kumagai et al.'s (2024) work on positive class distribution shift and Lu et al.'s (2022) advancements in importance weighting methods for domain adaptation and transfer learning.

Kumagai, A., Iwata, T., Takahashi, H., Nishiyama, T., and Fujiwara, Y. Auc maximization under positive distribution shift. In NeurIPS, 2024.
Lu, N., Zhang, T., Fang, T., Teshima, T., and Sugiyama, M. Rethinking importance weighting for transfer learning. In Federated and Transfer Learning, pp. 185–231. Springer, 2022.

**Theoretical Claims:**

The theoretical claims in the paper are both correct and rigorous.

---

> ### Author Rebuttal · Authors · 2025-03-28
>
> Thank you for your positive comments and constructive feedback.
> We added experimental results in the anonymous URL: https://anonymous.4open.science/r/icml25s-6C74/results.pdf
>
> > Weaknesses: (1) The paper lacks a clear explanation of the motivation for deriving the two estimators of the AUC risk on the test distribution under covariate shift.
>
> We have described the motivation for deriving the two AUC risk estimators in Section 4.2 and the fourth paragraph of Section 1.
> Specifically, under the covariate shift, existing AUC maximization methods cannot minimize the test AUC risk.
> To deal with this, we derived two AUC risk estimators that can approximate the test AUC risk.
> Since these estimators are calculated with different data for classifier learning (i.e., Eqs. (17) and (19)), we can use all available data $X ^{{\rm p}} _{{\rm tr}} \cup X _{{\rm tr}} \cup X _{{\rm te}}$ for classifier learning by using both estimators (i.e., Eq. (20)). We will clarify this.
>
> > there is no analysis of their error bounds.
>
> Thank you for the important feedback. We would like to work on analyzing the error bounds in the future.
>
> > (1) Has the issue of noise been considered? When the unlabeled data contains significant noise or redundant information, what impact would this have on the performance of the proposed AUC maximization method?
>
> Yes, we have considered the issue of noise.
> This is because unlabeled data can be regarded as noisy negative data since it contains a small number of positive data (noise).
> As class-prior $\pi$ becomes large (the noise in unlabeled data becomes large), the performance of our method tends to decrease, as described in Table 1. Nevertheless, our method outperformed the other methods.
>
> > (2) ... in real-world applications, such inconsistencies might also manifest as changes in class distributions (positive distribution shift) or shifts in class priors. In these inconsistent scenarios, can the proposed method still effectively perform AUC maximization?
>
> Thank you for the insightful question.
> Please see the answer to the last comment of Reviewer Gocn.
>
> > (1) The design in the paper relies on the sigmoid loss function. Would other loss functions be compatible with this solution approach?
>
> We can derive the same form of the loss function of Eqs. (17) and (19) when using symmetric loss functions as described in Lines 259--267.
> The symmetric functions include many popular losses, such as sigmoid, ramp, and unhinged losses.
>
> > (2) Why do the PAUC results in Table 1 perform particularly poorly, even being the worst in most cases? They are troublingly bad.
>
> PAUC strongly depends on the assumption that the negative-conditional density does not change between the training and testing to derive the loss function.
> However, the negative density changed significantly in our datasets.
> In addition, as noted in the PAUC paper, PAUC performs poorly when $\pi_{{\rm te}}$ is small due to its reliance on extracting positive data from unlabeled test data.
> For these reasons, PAUC did not work well in our setting.
> Note that we have confirmed that PAUC performs well in the setting used in the PAUC paper.
>
> > (3) The results on the SVHN dataset are consistently anomalous.
>
> Since each dataset has different properties, it is unsurprising that the trends of the results differ across datasets.
> Meanwhile, the SVHN is not the only data that shows different patterns.
> For example, although trPU, tePU, and CPU worked relatively well in MNIST and FashionMNIST, they performed poorly (AUC is about 0.5) in both SVHN and CIFAR10 in Table 1.
>
> > (4) Fig.2 shows the importance weight distribution on the FashionMNIST dataset when the positive class prior is 0.1. Does this result suggest that the method might overly rely on the test data during training? Furthermore, how do the distributions behave when the priors are set to 0.01/0.05?
>
> This result does not mean our method overly relies on the test data for classifier learning.
> This is because the (estimated) importance weights are used for only PU data in the training distribution in Eqs. (17) and (19).
>
> As described in the second paragraph of Section 5.1, the covariate shift was created so that the high- and low-density regions of the inputs were reversed between the training and testing.
> Since importance weights are defined as density-ratio $w({\bf x}) = p_{{\rm te}}({\bf x})/p_{{\rm tr}}({\bf x})$, test data would tend to have larger weights than training data.
> (This is also valid for the relative density-ratio in Eq. (21)).
> Figure 2 examined whether the estimated weights correctly reflect this characteristic.
> When the class-priors are $0.01$ or $0.05$, similar results are obtained in Figure A in the anonymous URL.
> We will clarify this.

---

### Official Review · Reviewer_Gocn · 2025-03-13

**Overall Recommendation:** 3

**Summary:**

The paper addresses the problem of maximizing the AUC in binary classification tasks under covariate shift, where the input distribution changes between training and test phases, but the conditional distribution of the class label given the input remains the same. The authors propose a novel method that leverages positive and unlabeled (PU) data from the training distribution and unlabeled data from the test distribution. They derive two estimators for the test AUC risk: one based on importance-weighted PU data from the training distribution, and another based on importance-weighted positive data from the training distribution and unlabeled data from the test distribution. The final loss function is a weighted sum of these two estimators. The authors also introduce a dynamic approach for importance weight estimation and classifier learning, which iteratively updates the importance weights and the classifier.

**Claims And Evidence:**

The claims made in the paper are generally supported by experimental results.

**Essential References Not Discussed:**

While the paper covers a broad range of related work, there are a few essential references that could provide additional context for the key contributions:
1. Nakajima S, Sugiyama M. Positive-unlabeled classification under class-prior shift: a prior-invariant approach based on density ratio estimation
2. Zhao Y, Xu Q, Jiang Y, et al. Dist-pu: Positive-unlabeled learning from a label distribution perspective
3. Jain S, White M, Radivojac P. Estimating the class prior and posterior from noisy positives and unlabeled data

**Experimental Designs Or Analyses:**

The scale of the adopted datasets is too small, e.g., only "50 positive and 3,000 unlabeled data in the training distribution and 3,000 unlabeled data in the test distribution". In fact, for benchmarks like MNIST/CIFAR10, the authors could use much more examples (e.g., 10000+ samples) for training and test. Besides, more recent AUC optimization methods and PU learning methods should be compared.

**Methods And Evaluation Criteria:**

The proposed methods are well-suited for the problem of AUC maximization under covariate shift.

**Other Comments Or Suggestions:**

None.

**Other Strengths And Weaknesses:**

Strengths:
1. The paper addresses a significant problem in PU learning under covariate shift, which is common in real-world applications.
2. The paper is generally well-written, explaining the proposed method, with detailed descriptions of each module.

Weaknesses:
1. The paper draws on the importance weighting framework from covariate shift literature (e.g., Sakai, T. and Shimizu, N. Covariate shift adaptation on learning from positive and unlabeled data. In AAAI, 2019) but adapts it for AUC maximization (e.g., Kumagai, A., Iwata, T., Takahashi, H., Nishiyama, T., and Fujiwara, Y. AUC maximization under positive distribution shift. In NeurIPS, 2024). In general, the novelty might be somewhat incremental.
2. The method assumes a specific type of distribution shift (covariate shift), which may limit its applicability in scenarios with other types of shifts.
3. The experimental design is insufficient to convincingly support the paper’s claims.

**Questions For Authors:**

None.

**Relation To Broader Scientific Literature:**

The key contributions of this paper are closely related to several areas of the broader scientific literature, including AUC maximization, covariate shift adaptation, and positive-unlabeled (PU) learning.

**Theoretical Claims:**

I checked the correctness of the proofs for the theoretical claims in the paper. Specifically, I reviewed the derivation of Lemma B.1 in Appendix B, which is used to prove Eq. (15). This proof is clear and correct.

---

> ### Author Rebuttal · Authors · 2025-03-28
>
> Thank you for your insightful comments and constructive feedback.
> We added experimental results in the anonymous URL: https://anonymous.4open.science/r/icml25s-6C74/results.pdf
>
> > The scale of the adopted datasets is too small, e.g., only "50 positive and 3,000 unlabeled data in the training distribution and 3,000 unlabeled data in the test distribution". In fact, for benchmarks like MNIST/CIFAR10, the authors could use much more examples (e.g., 10000+ samples) for training and test. Besides, more recent AUC optimization methods and PU learning methods should be compared.
>
> Thank you for the constructive suggestion. We performed the additional experiments on larger datasets.
> For each dataset, we used 100 positive and 9,000 unlabeled data in the training distribution and 9,000 unlabeled data in the test distribution for training.
> In addition, we included the recent PU learning method (PURA) [a] for comparison. Since it is designed for ordinary PU learning, it used PU data in the training distribution.
> Margin $\rho$ was selected from $\\{0.1,1,10\\}$ by validation data.
> The results are described in Table A of the anonymous URL.
> Our method outperformed the other methods.
> Since PURA does not consider distribution shift, it did not work well.
> We will include these results in the revised paper.
>
> [a] Positive-Unlabeled Learning with Label Distribution Alignment, TPAMI'23
>
> > While the paper covers a broad range of related work, there are a few essential references that could provide additional context for the key contributions:
>
> Thank you for sharing the papers.
> We briefly explain the difference between our work and these papers.
> The paper [1] proposed a PU learning method under class-prior shift. Unlike our method, it cannot deal with the covariate shift.
> The paper [2] proposed a PU learning method with label distribution consistency, but it cannot treat distribution shifts.
> The paper [3] proposed a PU learning method from noisy positive and unlabeled data. It cannot treat distribution shifts.
> In addition, all these methods are not designed for AUC maximization.
> We will include a more detailed discussion of these papers in Section 2.
>
> > The paper draws on the importance weighting framework from covariate shift literature (e.g., Sakai, T. and Shimizu, N. Covariate shift adaptation on learning from positive and unlabeled data. In AAAI, 2019) but adapts it for AUC maximization (e.g., Kumagai, A., Iwata, T., Takahashi, H., Nishiyama, T., and Fujiwara, Y. AUC maximization under positive distribution shift. In NeurIPS, 2024). In general, the novelty might be somewhat incremental.
>
> The novelty of our paper is mainly twofold.
> The first is to propose a novel and significant problem setting of AUC maximization under covariate shift with PU data.
> The second is to theoretically derive two novel AUC risk estimators to address this problem based on importance weighting, effectively utilizing the properties of AUC maximization and PU learning.
> Moreover, we also proposed to use the recently proposed dynamic importance weighting framework (Fang et al., 2020; 2023) for AUC maximization with PU data on complex models and data.
> As other reviewers acknowledged, we believe our method has sufficient technical novelty.
>
> > The method assumes a specific type of distribution shift (covariate shift), which may limit its applicability in scenarios with other types of shifts.
>
> As you mentioned, our paper assumes the covariate shift.
> This is because the covariate shift is the most common and important shift in practice (He et al., 2023).
> Note that unsupervised distribution shift adaptation methods require some assumption about the shift type since no supervision is available in the test distribution.
> Since many papers at top conferences focus on the covariate shift only [b, c, d, e, f], we believe that our work has sufficient contributions.
>
> [b] Test-time Adaptation for Regression by Subspace Alignment, ICLR'25
>
> [c] Robust importance weighting for covariate shift, AISTATS'20
>
> [d] Double-weighting for covariate shift adaptation, ICML'23
>
> [e] Adapting to continuous covariate shift via online density ratio estimation, NeurIPS'23
>
> [f] An information-theoretical approach to semi-supervised learning under covariate-shift, AISTATS'22
>
> Meanwhile, we additionally evaluated our method under a class-prior shift, in which the class-prior changes, but the class-conditional density remains the same.
> Table C in the anonymous URL shows the results.
> Our method empirically worked well.
> This would be because our method does not depend on class-priors and thus is relatively robust against the class-prior shift.
> We will include these results.

---

### Official Review · Reviewer_7MQt · 2025-03-14

**Overall Recommendation:** 4

**Summary:**

This paper proposes a new method for AUC maximization under covariate shift using positive-unlabeled (PU) data from the training distribution and unlabeled data from the test distribution. Given the challenges of estimating class priors of the training and test distribution, this paper theoretically derives two estimators for maximizing AUC such that they do not require the estimation of class priors. The paper also proposes an approach for importance weighting estimation for the AUC risk estimators. Empirical experiments on image and tabular datasets show overall performance improvement over existing approaches.

**Claims And Evidence:**

The main claim of the paper – that learning from PU training data and unlabeled test while adapting to covariate shift, using their AUC risk estimators improves on the state-of-the-art – is supported by theoretical proofs and experimental results.

**Essential References Not Discussed:**

The related works sufficiently describes relevant literature on this topic. The proposed method is compared against a large set of existing approaches published in recent years.

**Experimental Designs Or Analyses:**

Covariate shift is simulated using standardized approaches from existing literature. Paper takes appropriate steps to for fair comparison among different approaches. Statistical significance tests confirm performance improvement across different datasets and class priors.

**Methods And Evaluation Criteria:**

Proposed method, the methods which are included for comparison, and evaluation on image and tabular benchmark datasets makes sense for this problem.

**Other Comments Or Suggestions:**

N/A

**Other Strengths And Weaknesses:**

N/A

**Questions For Authors:**

N/A

**Relation To Broader Scientific Literature:**

The key contributions of the paper build on existing approaches in this domain but highlight the importance of specific parts of their proposed approach (such as recommending joint estimation of importance weighting during classifier learning over two-step importance weighting) which allow this approach to outperform existing state-of-the-art models.

**Theoretical Claims:**

Theoretical derivation of the two AUC risk estimators, approach to combine them, and approach to estimate importance weights seem to be correct. The steps described in the paper are detailed and make good use of the page limits.

---

> ### Author Rebuttal · Authors · 2025-03-28
>
> We appreciate your positive comments on our paper.

---

### Official Review · Reviewer_Su7H · 2025-03-14

**Overall Recommendation:** 4

**Summary:**

The paper aims to optimize the AUC under a covariate shift, i.e., when the test distribution of inputs differs from the training distribution. Considering the difficulty of collecting negative examples, this work focuses on the positive-unlabelled (PU) setting. To solve this problem, the paper first proposes two importance-weight-based risk estimators that depend only on PU training and unlabelled test data. A similar technique estimates the importance weights with a learnable model. Empirical studies on MNIST, FashionMNIST, SVHN, and CIFAR10 are conducted to validate the proposed method.

**Claims And Evidence:**

The claims are overall clear and well-supported:

**Claim 1.** New setting: The paper proposes to optimize AUC under the covariate shift with PU data.

**Evidence:** Related work in Section 2 indicates that such a setting is different from previous work and reasonable.

**Claim 2.** New methods: Two estimators of the AUC risk are proposed, which can be used to learn a score function without class priors; a dynamic method for predicting importance weights is provided.

**Evidence:** According to Eq. (17) and Eq. (19), the proposed estimators are unbiased and do not require the class priors. Besides, Eq. (23) provides a loss to fit a bounded extension of the importance weights.

**Claim 3.** Superior experimental performance.

**Evidence:** Table 1 shows that the proposed method outperforms most of the competitors in various datasets.

**Essential References Not Discussed:**

N/A

**Experimental Designs Or Analyses:**

The experiments could be further improved in the following aspects:
1. Both datasets and models (a few layers) are small-scale. It should be validated whether the proposed method is scalable to larger datasets or backbones (e.g., ResNet or DenseNet).
2. This work focuses on the covariate shift, i.e., **the input distributions are different** ($p\_{tr}(x) \neq p\_{te}(x)$). However, as described in lines 315~328, the training data and test data are sampled under different **class priors** (categories of the images). A sampling strategy based on the inputs (e.g., style, noise, color) instead of the classes would be more reasonable.
3. Details of the competitors (teAUC, trteAUC, UDAUC) are missing. To ensure fairness, how these methods utilize different data could be clarified.
4. Some results seem counter-intuitive. PAUC only achieves about 0.23~0.5 under most settings, but even a random guessing achieves 0.5 of AUC. Please provide more analyses and results, such as the AUC in the training distributions. Another issue is the effect of class prior $\pi$. As $\pi$ varies from 0.01 to 0.1, the tasks become more manageable with more positive examples, but the test AUC drops for most methods.
5. The hyperparameter $\alpha$ controls the trade-off between the upper bound and the approximation error in the importance weight estimation. Therefore, sensitive analysis on $\alpha$ should be provided.
6. According to Appendix D, the learning rate is set to $10^{-4}$ for all methods, which might be suboptimal for some competitors. A common strategy is searching for the best learning rate for each method.

**Methods And Evaluation Criteria:**

The proposed method makes sense since it can optimize AUC using only PU data without class priors. However, the benchmark datasets could be further improved: only small-scale datasets (resolution $< 32 \times 32$ and about tens of thousands of images) are used to validate the effectiveness. The results would be more convincing if larger benchmarks were used, e.g., Tiny-ImageNet-LT or iNaturalist.

**Other Comments Or Suggestions:**

1. Some equations are overwidth (e.g., Eq. (6), Eq. (7), Eq. (14)).
2. An abused notation $m$ is the index of unlabelled data before Eq. (20) and indicates the weights estimator in Eq. (22).

**Other Strengths And Weaknesses:**

### **Other Strengths**
1. Some real-world applications, such as cyber security and medical care, support the idea of jointly considering AUC optimization and covariate shift.
2. The presentation is clear and easy to understand overall.

### **Other Weaknesses**
1. The main concern is the soundness of the experiments (see Experimental Designs Or Analyses). This work would be more attractive if the experiment issues were addressed.
2.  The organization could be further improved. For example, the main paper presents too many detailed proofs. Although it is clear for readers who spend more time reading this paper, it is hard to find the main conclusions at first glance.

**Questions For Authors:**

1. A key to removing the positive-positive loss in Eq. (15) is using a symmetric function ($\sigma(z)+\sigma(-z)=1$). However, such a surrogate loss might suffer from a gradient vanishing problem if the predicted scores are near $0$ or $1$. Is it possible to practically use other surrogate losses, such as hinge or square loss?
2. How does the proposed method perform without a covariate shift? In this case, is the proposed method still comparable to previous PU & AUC optimization methods?

**Relation To Broader Scientific Literature:**

The paper extends AUC optimization, PU learning, and covariate shift. These areas are well studied, but the paper is the first work to jointly consider these problems. The main idea follows the mainstream covariate shift methods and develops some novel techniques. It could be applied to cyber security and medical care.

**Theoretical Claims:**

I have checked and confirmed the correctness of the proofs.

---

> ### Author Rebuttal · Authors · 2025-03-28
>
> Thank you for your positive and constructive comments.
> We will revise the paper according to your suggestion regarding the structure and notation.
> We added experimental results in the anonymous URL: https://anonymous.4open.science/r/icml25s-6C74/results.pdf
>
> > the benchmark datasets could be further improved: only small-scale datasets are used to validate the effectiveness.
>
> We performed the additional experiments on larger datasets.
> Our method outperformed the others in Table A of the anonymous URL.
> Please see the answer to the first comment of Reviewer Gocn for details.
>
> > This work focuses on the covariate shift, i.e., the input distributions are different. However, as described in lines 315~328, the training data and test data are sampled under different class priors (categories of the images). A sampling strategy based on the inputs (e.g., style, noise, color) instead of the classes would be more reasonable.
>
> Thank you for the constructive comment.
> As described in Line 317, we created the covariate shift by using the sampling strategy of the AISTATS paper of covariate shift adaptation (Aminian et al., 2022).
> Specifically, we first constructed positive and negative classes by partitioning the dataset's original classes into two groups.
> Then, we created $p_{{\rm tr}} ({\bf x}) \neq p_{{\rm te}} ({\bf x})$ by reversing the high- and low-density input regions based on the original classes between the training and testing.
> In our setting, training and test class-priors were the same ($\pi_{{\rm tr}} = \pi_{{\rm te}}$).
> Also, we used the epsilon dataset in Appendix E.3, where the covariate shift was created based on the inputs (the distance between the feature vectors) as in (Sakai & Shimizu, 2019).
> We would like to use the suggested strategy in future.
>
> > Details of the competitors (teAUC, trteAUC, UDAUC) are missing.
>
> As described in the second paragraph of Section 5.2, the loss functions of teAUC and trteAUC are equivalent to Eq. (19) and Eq. (20) with $w({\bf x})=1$ for all ${\bf x}$ (i.e., they do not use importance weights), respectively.
> The loss function of UDAUC is a weighted sum of the loss of trAUC and the coral loss that is calculated from unlabeled training and test data.
> The coral loss is used to minimize the feature discrepancy.
> teAUC used positive training and unlabeled test data.
> trteAUC and UDAUC used positive training, unlabeled training data, and unlabeled test data as in our method.
> We will describe their specific loss functions in the revised paper.
>
> > Some results seem counter-intuitive. PAUC only achieves about 0.23~0.5 under most settings.
>
> Please see our response to Reviewer 9PVZ's comment '(2) Why do the PAUC results in Table 1 perform particularly poorly.'
>
> > As $\pi$ varies from 0.01 to 0.1, the tasks become more manageable with more positive examples, but the test AUC drops for most methods.
>
> AUC-based methods, except for PAUC, prefer small $\pi$.
> This is because they essentially use loss functions of the form, $\mathbb{E} _{{\bf x} ^{{\rm p}} \sim p ^{{\rm p}}} \mathbb{E} _{{\bf x} \sim p} \left[ f({\bf x} ^{{\rm p}}, {\bf x} ) \right]$, where $p ^{{\rm p}}$ and $p$ are positive and marginal densities.
> When $\pi$ is small, $p$ can be regarded as negative density $p ^{{\rm n}}$. In this case, the above loss function becomes the original AUC risk.
> Thus, these methods work well with small $\pi$.
> Since PAUC has a different form of loss, its trend was different.
> We will clarify this.
>
> > sensitive analysis on $\alpha$ should be provided.
>
> We have performed the sensitive analysis on $\alpha$ in Appendix E.2.
> Although the tendency of the results varied across datasets, our method with $\alpha = 0.5$ worked well.
>
> > the learning rate is set to $10^{-4}$ for all methods, which might be suboptimal for some competitors.
>
>  In our preliminary experiments, we tested learning rates of $10^{-3}$ and $10^{-4}$ and confirmed no significant difference in performance. Based on this, we used $10^{-4}$ in this paper. We will mention this.
>
> > A key to removing the positive-positive loss in Eq. (15) is using a symmetric function... Is it possible to practically use other surrogate losses, such as hinge or square loss?
>
> Yes, we can use non-symmetric losses, although the second term in Eq. (14) is not a constant as you mentioned.
> In this case, $\pi_{{\rm tr}}$ is necessary for training, but it can be estimated from PU data (Kumagai et al., 2024).
> Since the sigmoid function worked well in many PU learning and AUC maximization works, we also used it. We will discuss this in the revised paper.
>
> > How does the proposed method perform without a covariate shift?
>
> We additionally evaluated our method when there were no shifts.
> Table B of the anonymous URL shows the results.
> Since there were no shifts, we compared trPU and trAUC, which do not consider shifts.
> Our method and trAUC showed comparable results, indicating that our method robustly works well when no shift exists.
> We will include this result.

---

> > ### Comment · Reviewer_Su7H · 2025-04-02
> >
> > Thanks for your detailed reply and additional experiments! Most concerns are well-addressed, but the experiments could be further improved in the following perspectives:
> > 1. **Larger datasets:** I agree that the updated datasets have more images, and the performance is consistent with previous results. However, the resolution of these datasets is still low ($\le 32\times 32$). In real-world scenarios, the resolution is generally larger than $224 \times 224$ for most images, so the effectiveness of the proposed method should be verified on images with larger resolutions.
> > 2. **Other model architecture:** Perhaps due to my imprecise description, the author missed the transferability issue of the proposed method to other model structures. Deep models used in most applications are much more complex than the MLPs with a few layers. In addition, for images with larger resolutions, the performance and efficiency of MLPs will be significantly reduced, so it is necessary to verify the effectiveness with other architectures.
> > 3. **The learning rates:** I agree that the Adam optimizer might be insensitive towards the learning rate for some methods. However, according to our experience, some losses require a significantly different learning rate to achieve their best performance. It is highly suggested to test other learning rates from $10^{-6}$ to $10^{-1}$.
> >
> > Based on the above considerations, I prefer to keep the original rating due to the experimental issues. If the authors only have limited GPUs, conducting these experiments might take a few days. Therefore, it is understandable that these issues cannot be fully addressed at the discussion stage, but the reviewer still expects the authors to fix these deficiencies as much as possible.

---

> > > ### Author Response · Authors · 2025-04-09
> > >
> > > Thank you very much for your reply and questions.
> > >
> > > >3:  The learning rates: I agree that the Adam optimizer might be insensitive towards the learning rate for some methods. However, according to our experience, some losses require a significantly different learning rate to achieve their best performance. It is highly suggested to test other learning rates from $10^{-6}$ to $10^{-1}$.
> > >
> > > Thank you for the valuable suggestion.
> > > We agree on the importance of the learning rate.
> > > Thus, we have investigated the performance obtained by varying the learning rate within $\\{10^{-6}, \dots, 10^{-1} \\}$.
> > > The average test AUCs over different class-prior $\pi$ within $\\{ 0.01, 0.05, 0.1 \\}$ of our method and trAUC, which is the most basic baseline, are as follows:
> > >
> > > | Method | Dataset  | $10^{-6}$ | $10^{-5}$ | $10^{-4}$ | $10^{-3}$ | $10^{-2}$ | $10^{-1}$ |
> > > |--------|----------|------|------|------|------|------|------|
> > > | Ours   | MNIST    | 0.709  | 0.760  | 0.804  | 0.806  | 0.796  | 0.738  |
> > > | Ours   | FMNIST   | 0.854  | 0.914  | 0.907  | 0.873  | 0.874  | 0.789  |
> > > | Ours   | SVHN     | 0.507 | 0.532  | 0.689  | 0.682  | 0.669  | 0.564  |
> > > | Ours   | CIFAR10  | 0.810  | 0.897  | 0.886  | 0.884  | 0.789  | 0.644  |
> > > | trAUC  | MNIST    | 0.693  | 0.756  | 0.787  | 0.785  | 0.779  | 0.784  |
> > > | trAUC  | FMNIST   | 0.846  | 0.916  | 0.909  | 0.895  | 0.891  | 0.876  |
> > > | trAUC  | SVHN     | 0.500 | 0.503  | 0.547  | 0.553  | 0.537  | 0.533  |
> > > | trAUC  | CIFAR10  | 0.761  | 0.879  | 0.874  | 0.870  | 0.865  | 0.738  |
> > >
> > > As observed, the $10^{-4}$ value used in our paper consistently shows good performance across all datasets.
> > > We would like to include this result in the revised paper.
> > >
> > > > 1 and 2: Larger datasets and other model architecture.
> > >
> > > Thank you for the important suggestion.
> > > We are currently conducting experiments using the Food101 dataset [a] and the ResNet-18 model, which allows us to evaluate our method with a larger dataset and model. As you have already recognized, due to our limited computational resources and time constraints, we have only been able to complete a portion of the experiments. Therefore, we apologize that this is a report of our current progress.
> > >
> > > The Food101 dataset consists of image data from 101 food categories and is widely used for image classification tasks.
> > > The maximum length of each image is 512 pixels.
> > > We resized each image to $224 \times 224$ pixels.
> > > To create a binary classification problem, we divided the original 101 categories into sweets-related (positive) and main dish-related (negative) classes. Then, following the procedure described in Section 5.1, we split the original categories within each positive/negative into two groups, assigning the first half with smaller class indices to the first group and the remaining to the second group. We then created the covariate shift by using the group ratio of 9:1 for the training and 1:9 for the testing. Details of the group assignments will be included in the revised paper.
> > > We used 2,500 (200) positive and 25,000 (2,000) unlabeled training data and 25,000 (2,000) unlabeled test data for training (validation). We used 3,000 test data for evaluation.
> > > As for ResNet-18, we did not use pre-trained weights to purely investigate the performance with the given data.
> > > We set the learning rate of Adam to $10^{-4}$ based on some empirical tuning.
> > > The average test AUCs over four trials are as follows:
> > >
> > > |    $\pi$   | Ours | trAUC | teAUC | trteAUC | UDAUC |
> > > |-------|------|-------|-------|---------|---------|
> > > | 0.01 | 0.590  | 0.585   | 0.601   | 0.585  | 0.577 |
> > > | 0.1  | 0.585  | 0.578   | 0.554   | 0.578  | 0.590 |
> > > | avg    | 0.588  | 0.582   | 0.578   | 0.582  | 0.584 |
> > >
> > > Here, we compared trAUC, teAUC, trteAUC, and UDAUC because they showed good results on the other datasets in Table 1 and are AUC-based methods like ours, which makes it easier to know the effect of the importance weighting.
> > > Our method performed slightly better than these methods on average.
> > > The performance may be further improved through more meticulous hyperparameter tuning.
> > > We also want to include the results of the other methods in the revised paper.
> > >
> > > Finally, thank you again for your helpful feedback. It helped us clarify our method's characteristics and improve the paper's quality.
> > >
> > > [a] Food-101 -- Mining Discriminative Components with Random Forests, ECCV'14

---

### Decision · Program_Chairs · 2025-05-01

**Decision:**

Accept (poster)

**Comment:**

This paper introduces an innovative approach aimed at maximizing the AUC in binary PU classification tasks, particularly in the context of covariate shift. The proposed methods and estimators are underpinned by a robust theoretical framework and exhibit good performance. Following the rebuttal, all reviewers expressed a consistently positive stance toward the work. Therefore, I recommend accepting this paper. However, it is strongly advised that the authors incorporate all relevant details in their rebuttal in the final version